# TIMEPERCEIVER: An Encoder-Decoder Framework for Generalized Time-Series Forecasting

**Jaebin Lee**
Sungkyunkwan University
jaebin.lee@skku.edu

**Hankook Lee**
Sungkyunkwan University
hankook.lee@skku.edu

## Abstract

In machine learning, effective modeling requires a holistic consideration of how to encode inputs, make predictions (*i.e.*, decoding), and train the model. However, in time-series forecasting, prior work has predominantly focused on encoder design, often treating prediction and training as separate or secondary concerns. In this paper, we propose TIMEPERCEIVER, a unified encoder-decoder forecasting framework that is tightly aligned with an effective training strategy. To be specific, we first *generalize the forecasting task* to include diverse temporal prediction objectives such as extrapolation, interpolation, and imputation. Since this generalization requires handling input and target segments that are arbitrarily positioned along the temporal axis, we design a novel encoder-decoder architecture that can flexibly perceive and adapt to these varying positions. For encoding, we introduce a set of *latent bottleneck representations* that can interact with all input segments to jointly capture temporal and cross-channel dependencies. For decoding, we leverage *learnable queries* corresponding to target timestamps to effectively retrieve relevant information. Extensive experiments demonstrate that our framework consistently and significantly outperforms prior state-of-the-art baselines across a wide range of benchmark datasets. The code is available at https://github.com/efficient-learning-lab/TimePerceiver.

## 1 Introduction

*Time-series forecasting* is a fundamental task in machine learning, aiming to predict future events based on past observations. It is of practical importance, as it plays a crucial role in many real-world applications, including weather forecasting [1], electricity consumption forecasting [2], and traffic flow prediction [3]. Despite decades of rapid advances in machine learning, time-series forecasting remains a challenging problem due to complex temporal dependencies, non-linear patterns, domain variability, and other factors. In recent years, numerous deep learning approaches [4–18] have been proposed to improve forecasting accuracy, and it continues to be an active area of research.

One promising and popular research direction is to design a new neural network architecture for time-series data, such as Transformers [4–9], convolutional neural networks (CNNs) [11, 12], multi-layer perceptrons (MLPs) [13–15], and state space models (SSMs) [17, 18]. These architectures primarily focus on capturing temporal and channel (*i.e.*, variate) dependencies within input signals, and how to *encode* the input into a meaningful representation. The encoder architectures are often categorized into two groups: channel-independent encoders, which treat each variate separately and apply the same encoder across all variates, and channel-dependent encoders, which explicitly model interactions among variates. The channel-independent encoders are considered simple yet robust [19]; however, they fundamentally overlook cross-channel interactions, which can be critical for multivariate time-series forecasting. In contrast, the channel-dependent encoders [5, 6, 8] can inherently capture such cross-channel dependencies, but they often suffer from high computational

39th Conference on Neural Information Processing Systems (NeurIPS 2025).

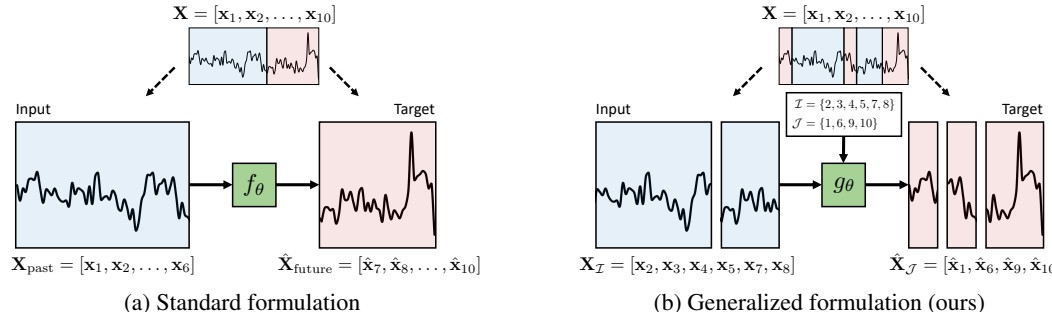

Figure 1: (a) The standard time-series forecasting task aims to predict only the future values from past observations. In contrast, (b) our generalized task formulation aims to predict not only the future, but also the past and missing values based on arbitrary contextual information.

cost and do not consistently yield significant improvements in forecasting accuracy over channel-independent baselines.

While the encoder architecture is undoubtedly a core component of time-series forecasting models, it is equally important to consider (*i*) how to accurately predict (*i.e.*, *decode*) future signals from the encoded representations of past signals, and (*ii*) how to effectively *train* the entire forecasting model. However, they often been studied independently, and little attention has been paid to how to effectively integrate them. For decoding, most prior works rely on a simple linear projection that directly predicts the future from the encoded representations. This design offers advantages in terms of simplicity and training efficiency, but may struggle to fully capture complex temporal structures. For training, inspired by BERT [20], masking-and-reconstruction tasks [4, 16] have been commonly adopted to pretrain encoders before supervised learning for time-series forecasting. In the pretraining stage, a subset of temporal segments from the time-series data is masked, and the encoder learns to reconstruct the masked portions. Despite its effectiveness, the self-supervised learning approach and the two-stage training (*i.e.*, pretraining-finetuning) strategy remain questionable whether it is truly aligned with architectural designs (*i.e.*, encoders and decoders) of time-series forecasting models.

**Contribution.** In this paper, we propose TIMEPERCEIVER, a unified framework that tightly integrates an encoder-decoder architecture with an effective training strategy tailored for time-series forecasting. Our key idea is to generalize the standard forecasting task—formulated as predicting future values from a sequence of past observations—into a broader formulation that encompasses extrapolation, interpolation, and imputation tasks along the temporal axis (see Figure 1). In this setting, the model learns to predict not only future values, but also past and missing values based on arbitrary contextual information. This generalized formulation enables the model to jointly learn temporal structures and predictive behavior in a single end-to-end training process, thereby fostering a deeper understanding of temporal dynamics and eliminating the need for a separate pretraining-finetuning pipeline.

To support our formulation, as illustrated in Figure 2, we design a novel attention-based encoder-decoder architecture that can flexibly handle arbitrary and potentially discontinuous temporal segments unlike conventional models that operate on fixed-length lookback windows and predict fixed-length forecasting horizons. Specifically, our encoder utilizes the cross-attention mechanism (*i*) to encode an arbitrary set of temporal segments into a fixed-size set of *latent bottleneck representations*, and (*ii*) to contextualize each segment by leveraging the bottleneck representations. This bottleneck process enables the encoder to efficiently capture both temporal and cross-channel dependencies. To enhance the quality of the bottleneck representations, we also apply the self-attention mechanism within the bottleneck set. After encoding input segments, our decoder generates predictions via cross-attention between the representations of the input segments and learnable queries that correspond to target timestamps. This allows the decoder to selectively retrieve relevant information of the input and to produce temporally-aware outputs. This design is naturally aligned with our learning objective, which includes the forecasting task as part of a broader set of temporal prediction tasks.

Through extensive experiments, our framework achieves state-of-the-art performance compared to recent baselines [4, 6–9, 11, 13, 14, 17] on standard benchmarks [1–3, 21, 22] (see Table 1). Notably, our model achieves **55** best and 17 second-best scores out of 80 settings, which are averaged over three different input lengths, demonstrating its strong overall performance. As a result, our framework records the best average rank with **1.375** in MSE and **1.550**, indicating its consistent superiority

over the baselines. We also conduct ablation studies to verify the effectiveness of each component (Section 4.2) and analyze attention maps to gain insights into how the model operates (Section 4.3).

Overall, our work emphasizes the importance of aligning architectural design with task formulation in time-series forecasting. We hope that this perspective encourages a shift from encoder-centric designs toward unified approaches more closely aligned with the core objectives of time-series forecasting.

## 2 Preliminaries

### 2.1 Problem Statement: Multivariate Time-Series Forecasting

In this paper, we aim to solve the task of *multivariate time-series forecasting*, which requires to predict future values of multiple variables based on past observations. To formally define the task, let $\mathbf{x}_t \in \mathbb{R}^C$ denote a multivariate observation at time step $t$, where $C$ is the number of variables (or channels). A multivariate time series of length $T$ can be written as a sequence $\mathbf{X} = [\mathbf{x}_1, \mathbf{x}_2, \ldots, \mathbf{x}_T] \in \mathbb{R}^{C \times T}$. Given a lookback window $\mathbf{X}_{\text{past}} = [\mathbf{x}_{t-L+1}, \ldots, \mathbf{x}_t]$ of length $L$, the goal of the task is to predict the future values $\mathbf{X}_{\text{future}} = [\mathbf{x}_{t+1}, \ldots, \mathbf{x}_{t+H}]$ over a forecasting horizon of length $H$.

Solving the task requires effectively capturing key characteristics of time-series data, such as temporal dependencies and cross-variable interactions. A common approach is to design a forecasting architecture $f_\theta$ that can model such properties, and to train it using a simple objective such as mean squared error between the predicted and ground-truth future values as follows (see Figure 1a):

$$\mathcal{L}(\theta; \mathbf{X}_{\text{past}}, \mathbf{X}_{\text{future}}) = \frac{1}{HC} \sum_{h=1}^{H} \|\hat{\mathbf{x}}_{t+h} - \mathbf{x}_{t+h}\|_2^2, \quad \hat{\mathbf{X}}_{\text{future}} = [\hat{\mathbf{x}}_{t+1}, \ldots, \hat{\mathbf{x}}_{t+H}] = f_\theta(\mathbf{X}_{\text{past}}). \quad (1)$$

This standard formulation, commonly used in many forecasting models and benchmarks [4, 6, 8, 22], assumes that both the lookback window and the forecasting horizon are of fixed length and continuous.

### 2.2 Attention Mechanism

In this section, we formally describe the attention mechanism [23], which plays an important role in our TIMEPERCEIVER framework. It is designed to dynamically capture dependencies between input tokens by computing their contextual relevance through learned similarity scores. Specifically, given queries $\mathbf{Q} \in \mathbb{R}^{N \times d}$, keys $\mathbf{K} \in \mathbb{R}^{M \times d}$, and values $\mathbf{V} \in \mathbb{R}^{M \times d}$, the attention output is computed as:

$$\texttt{Attention}(\mathbf{Q}, \mathbf{K}, \mathbf{V}) = \texttt{Softmax}\left(\frac{\mathbf{Q}\mathbf{K}^\top}{\sqrt{d}}\right)\mathbf{V}. \quad (2)$$

This has recently become a standard architectural component across various domains, including including NLP [23], vision [24], tabular data [23], and time-series forecasting [4–7]. A common design for modular blocks combines attention with skip connections and feedforward networks (FFNs):

$$\mathbf{H} = \mathbf{U} + \texttt{Attention}(\mathbf{U}, \mathbf{Z}, \mathbf{Z}), \quad (3)$$

$$\texttt{AttnBlock}(\mathbf{U}, \mathbf{Z}) = \mathbf{H} + \texttt{FFN}(\mathbf{H}), \quad (4)$$

where $\mathbf{U} \in \mathbb{R}^{N \times d}$ and $\mathbf{Z} \in \mathbb{R}^{M \times d}$ denote input and context tokens, respectively. For simplicity, we omit learnable parameters, layer normalization [25], and multi-head attention [23]. When $\mathbf{U} = \mathbf{Z}$, the block corresponds to self-attention; otherwise, it performs cross-attention, allowing input tokens to attend to an external context.

In the context of time-series forecasting, attention-based models adopt various tokenization strategies to capture temporal and multivariate patterns. A common approach is to divide the input time series into contiguous fixed-length temporal segments, or patches, and treat each patch as a token [4, 5]. Alternatively, DeformableTST [7] selectively samples important time steps and treats them as tokens to capture temporal patterns, while iTransformer [6] represents the entire time series of each channel as a single token. In this work, we simply adopt the common approach.

## 3 Methodology

In this work, we propose TIMEPERCEIVER, a unified encoder-decoder framework for generalized time-series forecasting. Our framework is based on a generalized formulation of the forecasting task

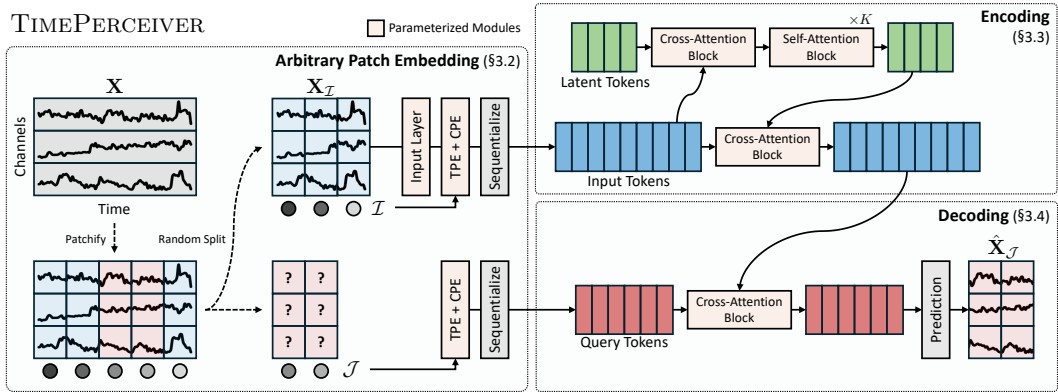

Figure 2: An overview of our TIMEPERCEIVER framework.

(Section 3.1 and 3.2) and consists of two main components tailored to the formulation: an encoder that jointly captures complex temporal and channel dependencies (Section 3.3), and a decoder that selects queries corresponding to target segments and retrieves the relevant context (Section 3.4). Our overall framework is illustrated in Figure 2.

## 3.1 A Generalized Formulation of Time-series Forecasting

While the formulation described in Section 2.1 explicitly follows the forecasting task definition, it inherently assumes a unidirectional temporal flow (*i.e.*, from past to future), which may limit the ability to fully capture the underlying temporal dynamics. To address this limitation, we propose a *generalized formulation* that extends the forecasting task to allow flexible conditioning on arbitrary temporal segments, enabling the model to learn a more comprehensive understanding of the temporal dynamics. To define our generalized formulation, we begin by introducing a notation for a time series of arbitrary temporal segments. Given a multivariate time series $\mathbf{X} = [\mathbf{x}_1, \mathbf{x}_2, \ldots, \mathbf{x}_T]$ and a set of time indices $\mathcal{I} = \{i_1, i_2, \ldots, i_{|\mathcal{I}|}\} \subseteq \{1, \ldots, T\}$, we define the corresponding series $\mathbf{X}_{\mathcal{I}}$ as follows:

$$\mathbf{X}_{\mathcal{I}} := [\mathbf{x}_i : i \in \mathcal{I}] = [\mathbf{x}_{i_1}, \ldots, \mathbf{x}_{i_{|\mathcal{I}|}}] \in \mathbb{R}^{C \times |\mathcal{I}|}. \tag{5}$$

Now, the generalized forecasting task can be defined by specifying a subset of time indices $\mathcal{I}$ as the input context and treating the remaining indices $\mathcal{J} = \{1, \ldots, T\} \setminus \mathcal{I}$ as prediction targets. In this task, a generalized forecasting model $g_\theta$ is expected to operate on any choice of $\mathcal{I}$ and $\mathcal{J}$, and to predict the values of the target indices as follows (see Figure 1b):

$$\hat{\mathbf{X}}_{\mathcal{J}} = g_\theta(\mathbf{X}_{\mathcal{I}}, \mathcal{I}, \mathcal{J}). \tag{6}$$

The model is trained to minimize the mean squared error over the target indices $\mathcal{J}$, formulated as:

$$\mathcal{L}(\theta; \mathbf{X}, \mathcal{I}, \mathcal{J}) = \frac{1}{|\mathcal{J}|C} \sum_{j \in \mathcal{J}} \|\hat{\mathbf{x}}_j - \mathbf{x}_j\|_2^2, \quad \hat{\mathbf{X}}_{\mathcal{J}} = [\hat{\mathbf{x}}_j : j \in \mathcal{J}] = g_\theta(\mathbf{X}_{\mathcal{I}}, \mathcal{I}, \mathcal{J}). \tag{7}$$

This generalized formulation allows both the input and target time indices to be arbitrary and non-contiguous, thereby encompassing extrapolation, interpolation, and imputation tasks and capturing more complex temporal patterns. For example, the standard model $f_\theta$ can be seen as its special case:

$$f_\theta(\mathbf{X}_{\text{past}}) = g_\theta(\mathbf{X}_{\text{past}}, \{t - L + 1, \ldots, t\}, \{t + 1, \ldots, t + H\}). \tag{8}$$

## 3.2 Constructing Arbitrary Patch Embeddings

Leveraging patch-based representations has recently become an important design choice in time-series forecasting models [4] inspired by the success of the Vision Transformer [24]. Motivated by this trend, we extend our formulation in Section 3.1 to its patch-based version. To this end, given a multivariate time series $\mathbf{X} \in \mathbb{R}^{C \times T}$, we first divide the time index set $\{1, 2, \ldots, T\}$ into $N$ disjoint subsets $\mathcal{P}_1, \mathcal{P}_2, \ldots, \mathcal{P}_N$ where $\mathcal{P}_i = \{(i - 1)P + 1, \ldots, iP\}$ and $P = T/N$ is the patch length. Here, we assume $T$ is divisible by $N$ for notational simplicity. This results in $N$ fixed-length and non-overlapping *patches* along the temporal axis, denoted as $\mathbf{X}_{\mathcal{P}_1}, \ldots, \mathbf{X}_{\mathcal{P}_N}$, where each patch is defined

as $\mathbf{X}_{\mathcal{P}_i} = [\mathbf{x}_t \in \mathbb{R}^C : t \in \mathcal{P}_i] \in \mathbb{R}^{C \times P}$. For our task formulation, we consider an arbitrary set of patch indices $\mathcal{I}_{\text{patch}} \subset \{1, 2, \ldots, N\}$ and define the remaining set as $\mathcal{J}_{\text{patch}} = \{1, \ldots, N\} \setminus \mathcal{I}_{\text{patch}}$. This corresponds to selecting input and target time indices as $\mathcal{I} = \bigcup_{i \in \mathcal{I}_{\text{patch}}} \mathcal{P}_i$ and $\mathcal{J} = \bigcup_{j \in \mathcal{J}_{\text{patch}}} \mathcal{P}_j$, respectively. Given the lookback window length $L$ in the forecasting task, we set the number of selected input patches as $|\mathcal{I}_{\text{patch}}| = L/P$, *i.e.*, $|\mathcal{I}| = L$.

To encode the structural information of patches $\mathbf{X}_{\mathcal{P}_1}, \ldots, \mathbf{X}_{\mathcal{P}_N}$, we introduce two learnable positional embeddings: (*i*) temporal positional embedding (TPE), denoted as $\mathbf{E}^{\text{temporal}} \in \mathbb{R}^{N \times D}$, which encodes the temporal location of each patch, and (*ii*) channel positional embedding (CPE), denoted as $\mathbf{E}^{\text{channel}} \in \mathbb{R}^{C \times D}$, which represents the identity of each channel. Using these positional embeddings, we construct the input patch embedding $\mathbf{H}^{(0)}$ and the query patch embedding $\mathbf{Q}^{(0)}$ as follows:

$$\mathbf{H}^{(0)}_{c,i} = \mathbf{X}_{\mathcal{P}_i,c} \mathbf{W}_{\text{input}} + \mathbf{E}^{\text{channel}}_c + \mathbf{E}^{\text{temporal}}_i \in \mathbb{R}^D, \qquad \forall i \in \mathcal{I}_{\text{patch}}, \ \forall c \in \{1, \ldots, C\}, \qquad (9)$$

$$\mathbf{Q}^{(0)}_{c,j} = \mathbf{E}^{\text{channel}}_c + \mathbf{E}^{\text{temporal}}_j \in \mathbb{R}^D, \qquad \forall j \in \mathcal{J}_{\text{patch}}, \ \forall c \in \{1, \ldots, C\}, \qquad (10)$$

where $\mathbf{X}_{\mathcal{P}_i,c} \in \mathbb{R}^P$ denotes the raw values of the $i$-th patch for channel $c$, and $\mathbf{W}_{\text{input}} \in \mathbb{R}^{P \times D}$ is a learnable input projection matrix. Note that $\mathbf{H}^{(0)}$ and $\mathbf{Q}^{(0)}$ are treated as sequences of embedding vectors in Section 3.3 and 3.4, *i.e.*, $\mathbf{H}^{(0)} \in \mathbb{R}^{(C|\mathcal{I}_{\text{patch}}|) \times D}$ and $\mathbf{Q}^{(0)} \in \mathbb{R}^{(C|\mathcal{J}_{\text{patch}}|) \times D}$.

## 3.3 Encoding with Latent Bottleneck

Given the input patch embeddings $\mathbf{H}^{(0)}$ constructed in Section 3.2, our encoder aims to efficiently capture both temporal and cross-channel dependencies within the embeddings for time-series forecasting. To this end, we propose *latent bottleneck*, a two-stage mechanism in which input tokens are first compressed into a fixed number of latent tokens and then projected back to the input tokens. This design enables efficient and selective modeling of key dependencies while avoiding the computational overhead of full attention.

We now formally describe our attention-based encoder based on the latent bottleneck mechanism. To this end, we introduce a set of learnable latent tokens $\mathbf{Z}^{(0)} \in \mathbb{R}^{M \times D}$ where $M$ is the number of latent tokens. Since these tokens can adaptively model key dependencies, $M$ can be significantly smaller than the number of input tokens $C|\mathcal{I}_{\text{patch}}|$. Staring from $\mathbf{Z}^{(0)}$, the latent bottleneck operates as follows:

1. The latent tokens attend to the input to collect contextual information:
$$\mathbf{Z}^{(1)} = \texttt{AttnBlock}(\mathbf{Z}^{(0)}, \mathbf{H}^{(0)}, \mathbf{H}^{(0)}) \in \mathbb{R}^{M \times D}.$$

2. The latent tokens are refined via $K$ self-attention layers:
$$\mathbf{Z}^{(k+1)} = \texttt{AttnBlock}(\mathbf{Z}^{(k)}, \mathbf{Z}^{(k)}, \mathbf{Z}^{(k)}), \quad \forall k = 1, \ldots, K.$$

3. The updated latent tokens are used to update the input tokens:
$$\mathbf{H}^{(1)} = \texttt{AttnBlock}(\mathbf{H}^{(0)}, \mathbf{Z}^{(K+1)}, \mathbf{Z}^{(K+1)}) \in \mathbb{R}^{(C|\mathcal{I}_{\text{patch}}|) \times D}.$$

This design acts as an efficient attention bottleneck that significantly reduces computation from $\mathcal{O}(N^2)$ in full attention to $\mathcal{O}(NM)$ while selectively preserving informative patterns across both temporal and channel dimensions. We provide further ablation studies on the encoding mechanism and the latent bottleneck in Section 4.2 and Appendix G.

## 3.4 Decoding via Querying Target Patches

Based on the query patch embeddings $\mathbf{Q}^{(0)}$ described in Section 3.2 and the encoded input patch embeddings $\mathbf{H}^{(1)}$ from Section 3.3, our decoder is designed to generate predictions $\hat{\mathbf{X}}_{\mathcal{P}_j}$ for target patches $\mathcal{P}_j$, where $j \in \mathcal{J}_{\text{patch}}$. Note that $\mathbf{Q}^{(0)} \in \mathbb{R}^{(C|\mathcal{J}_{\text{patch}}|) \times D}$ consists of $C \times |\mathcal{J}_{\text{patch}}|$ query tokens where each token $\mathbf{Q}^{(0)}_{c,j} = \mathbb{E}^{\text{channel}}_c + \mathbb{E}^{\text{temporal}}_j \in \mathbb{R}^D$ corresponds to the query vector for channel $c$ and patch $\mathcal{P}_j$, as defined in Section 3.2.

To retrieve relevant information from the input for each query, we apply cross-attention using $\mathbf{Q}^{(0)}$ as queries and $\mathbf{H}^{(1)}$ as keys and values:

$$\mathbf{Q}^{(1)} = \texttt{AttnBlock}(\mathbf{Q}^{(0)}, \mathbf{H}^{(1)}, \mathbf{H}^{(1)}) \in \mathbb{R}^{(C|\mathcal{J}_{\text{patch}}|) \times D}.$$

Table 1: Multivariate long-term forecasting results compared with baseline models. **Bold** indicates the best result, and underlined denotes the second best. The results are averaged over input lengths $L \in \{96, 384, 768\}$ and prediction lengths $H \in \{96, 192, 336, 720\}$. Rank indicates the average position of each model in MSE and MAE across all prediction lengths. Results for each individual input length are provided in Appendix E.

| Models | | **TimePerceiver** | | DeformableTST | | CARD | | CATS | | PatchTST | | iTransformer | | S-Mamba | | DLinear | | TimesNet | | TiDE | |
|---|---|---|---|---|---|---|---|---|---|---|---|---|---|---|---|---|---|---|---|---|---|
| Metric | | MSE | MAE | MSE | MAE | MSE | MAE | MSE | MAE | MSE | MAE | MSE | MAE | MSE | MAE | MSE | MAE | MSE | MAE | MSE | MAE |
| Weather | 96 | **0.147** | **0.199** | 0.155 | 0.203 | 0.156 | 0.208 | 0.150 | 0.201 | 0.158 | 0.207 | 0.166 | 0.214 | 0.162 | 0.212 | 0.178 | 0.238 | 0.169 | 0.221 | 0.183 | 0.238 |
| | 192 | **0.193** | **0.242** | 0.200 | 0.244 | 0.214 | 0.258 | 0.196 | 0.244 | 0.204 | 0.248 | 0.210 | 0.253 | 0.209 | 0.254 | 0.221 | 0.278 | 0.237 | 0.278 | 0.225 | 0.274 |
| | 336 | **0.246** | **0.283** | 0.252 | 0.284 | 0.263 | 0.295 | 0.249 | 0.284 | 0.255 | 0.286 | 0.263 | 0.293 | 0.266 | 0.296 | 0.268 | 0.319 | 0.292 | 0.319 | 0.270 | 0.309 |
| | 720 | 0.323 | 0.338 | 0.325 | **0.335** | 0.354 | 0.351 | **0.322** | **0.335** | 0.328 | 0.336 | 0.335 | 0.343 | 0.334 | 0.342 | 0.328 | 0.366 | 0.385 | 0.377 | 0.335 | 0.357 |
| | Avg | **0.227** | **0.265** | 0.233 | 0.266 | 0.247 | 0.278 | 0.229 | 0.266 | 0.236 | 0.269 | 0.244 | 0.276 | 0.243 | 0.276 | 0.249 | 0.300 | 0.271 | 0.299 | 0.254 | 0.295 |
| Solar | 96 | **0.173** | **0.221** | 0.177 | 0.237 | 0.194 | 0.257 | 0.184 | 0.228 | 0.207 | 0.283 | 0.188 | 0.242 | 0.191 | 0.242 | 0.232 | 0.322 | 0.229 | 0.294 | 0.246 | 0.324 |
| | 192 | **0.195** | **0.238** | 0.199 | 0.254 | 0.229 | 0.279 | 0.210 | 0.243 | 0.231 | 0.304 | 0.212 | 0.263 | 0.219 | 0.268 | 0.258 | 0.341 | 0.266 | 0.318 | 0.271 | 0.351 |
| | 336 | **0.206** | **0.248** | 0.208 | 0.263 | 0.239 | 0.289 | 0.218 | 0.253 | 0.249 | 0.312 | 0.226 | 0.275 | 0.229 | 0.279 | 0.280 | 0.355 | 0.275 | 0.337 | 0.291 | 0.367 |
| | 720 | 0.217 | **0.259** | **0.214** | 0.266 | 0.251 | 0.301 | 0.233 | 0.261 | 0.250 | 0.314 | 0.232 | 0.282 | 0.236 | 0.284 | 0.291 | 0.360 | 0.276 | 0.329 | 0.300 | 0.364 |
| | Avg | **0.198** | **0.241** | 0.199 | 0.255 | 0.228 | 0.282 | 0.211 | 0.247 | 0.234 | 0.303 | 0.214 | 0.266 | 0.219 | 0.268 | 0.266 | 0.345 | 0.261 | 0.319 | 0.277 | 0.351 |
| Electricity | 96 | **0.128** | **0.223** | 0.141 | 0.240 | 0.140 | 0.236 | 0.134 | 0.228 | 0.148 | 0.246 | 0.143 | 0.240 | 0.136 | 0.232 | 0.164 | 0.258 | 0.188 | 0.294 | 0.180 | 0.282 |
| | 192 | **0.146** | **0.240** | 0.155 | 0.254 | 0.161 | 0.257 | 0.153 | 0.243 | 0.163 | 0.256 | 0.160 | 0.256 | 0.157 | 0.253 | 0.173 | 0.268 | 0.197 | 0.303 | 0.186 | 0.287 |
| | 336 | **0.165** | **0.259** | 0.172 | 0.271 | 0.181 | 0.279 | 0.171 | 0.262 | 0.180 | 0.274 | 0.177 | 0.274 | 0.173 | 0.270 | 0.188 | 0.285 | 0.229 | 0.327 | 0.202 | 0.301 |
| | 720 | 0.204 | 0.293 | 0.208 | 0.303 | 0.212 | 0.304 | 0.204 | **0.292** | 0.218 | 0.306 | 0.220 | 0.313 | **0.198** | 0.293 | 0.223 | 0.318 | 0.239 | 0.337 | 0.236 | 0.328 |
| | Avg | **0.161** | **0.254** | 0.169 | 0.267 | 0.174 | 0.269 | 0.166 | 0.257 | 0.177 | 0.271 | 0.175 | 0.271 | 0.166 | 0.262 | 0.187 | 0.282 | 0.213 | 0.315 | 0.201 | 0.300 |
| Traffic | 96 | 0.381 | 0.263 | 0.378 | 0.265 | 0.399 | 0.279 | 0.379 | 0.255 | 0.407 | 0.280 | 0.389 | 0.286 | **0.357** | **0.254** | 0.493 | 0.329 | 0.571 | 0.310 | 0.582 | 0.400 |
| | 192 | 0.394 | 0.265 | 0.401 | 0.272 | 0.415 | 0.283 | 0.400 | 0.266 | 0.417 | 0.283 | 0.410 | 0.298 | **0.371** | **0.262** | 0.482 | 0.323 | 0.584 | 0.316 | 0.569 | 0.397 |
| | 336 | 0.409 | 0.273 | 0.413 | 0.282 | 0.429 | 0.293 | 0.413 | 0.271 | 0.430 | 0.289 | 0.428 | 0.307 | **0.382** | **0.270** | 0.493 | 0.328 | 0.597 | 0.317 | 0.581 | 0.404 |
| | 720 | 0.445 | 0.293 | 0.448 | 0.300 | 0.461 | 0.308 | 0.448 | 0.294 | 0.465 | 0.307 | 0.468 | 0.331 | **0.438** | 0.295 | 0.530 | 0.350 | 0.619 | 0.330 | 0.583 | 0.405 |
| | Avg | 0.407 | 0.273 | 0.410 | 0.280 | 0.426 | 0.291 | 0.411 | 0.272 | 0.430 | 0.290 | 0.424 | 0.306 | **0.387** | **0.270** | 0.499 | 0.333 | 0.593 | 0.318 | 0.579 | 0.401 |
| ETTh1 | 96 | **0.364** | **0.392** | 0.370 | 0.399 | 0.384 | 0.406 | 0.375 | 0.402 | 0.390 | 0.410 | 0.400 | 0.420 | 0.392 | 0.416 | 0.377 | 0.400 | 0.436 | 0.438 | 0.459 | 0.459 |
| | 192 | **0.402** | **0.417** | 0.415 | 0.425 | 0.421 | 0.426 | 0.419 | 0.432 | 0.429 | 0.433 | 0.441 | 0.446 | 0.437 | 0.444 | 0.418 | 0.426 | 0.481 | 0.467 | 0.497 | 0.481 |
| | 336 | **0.425** | **0.431** | 0.411 | 0.426 | 0.451 | 0.445 | 0.443 | 0.445 | 0.458 | 0.452 | 0.472 | 0.466 | 0.470 | 0.467 | 0.481 | 0.451 | 0.561 | 0.503 | 0.525 | 0.499 |
| | 720 | **0.450** | **0.461** | 0.456 | 0.469 | 0.463 | 0.473 | 0.476 | 0.470 | 0.473 | 0.481 | 0.533 | 0.520 | 0.514 | 0.508 | 0.495 | 0.508 | 0.615 | 0.533 | 0.541 | 0.529 |
| | Avg | **0.410** | **0.426** | 0.413 | 0.430 | 0.430 | 0.438 | 0.428 | 0.438 | 0.438 | 0.444 | 0.461 | 0.463 | 0.445 | 0.459 | 0.436 | 0.447 | 0.524 | 0.485 | 0.505 | 0.492 |
| ETTh2 | 96 | 0.278 | 0.336 | **0.274** | **0.334** | 0.283 | 0.339 | 0.282 | 0.342 | 0.284 | 0.341 | 0.303 | 0.346 | 0.298 | 0.354 | 0.321 | 0.380 | 0.373 | 0.405 | 0.378 | 0.433 |
| | 192 | 0.347 | 0.388 | **0.334** | **0.374** | 0.355 | 0.384 | 0.348 | 0.385 | 0.354 | 0.384 | 0.396 | 0.415 | 0.371 | 0.400 | 0.436 | 0.451 | 0.416 | 0.438 | 0.468 | 0.485 |
| | 336 | 0.354 | 0.407 | **0.328** | **0.377** | 0.382 | 0.409 | 0.372 | 0.405 | 0.378 | 0.408 | 0.427 | 0.438 | 0.403 | 0.425 | 0.533 | 0.508 | 0.469 | 0.469 | 0.524 | 0.522 |
| | 720 | **0.395** | 0.437 | 0.406 | 0.437 | 0.401 | **0.431** | 0.418 | 0.448 | 0.409 | 0.439 | 0.432 | 0.454 | 0.423 | 0.448 | 0.847 | 0.656 | 0.484 | 0.482 | 0.624 | 0.579 |
| | Avg | 0.344 | 0.392 | **0.336** | **0.381** | 0.355 | 0.391 | 0.355 | 0.395 | 0.356 | 0.394 | 0.390 | 0.417 | 0.374 | 0.407 | 0.535 | 0.499 | 0.436 | 0.450 | 0.499 | 0.505 |
| ETTm1 | 96 | **0.291** | **0.339** | 0.300 | 0.350 | 0.306 | 0.349 | 0.296 | 0.342 | 0.305 | 0.354 | 0.321 | 0.368 | 0.317 | 0.365 | 0.318 | 0.356 | 0.351 | 0.383 | 0.331 | 0.367 |
| | 192 | **0.329** | **0.365** | 0.338 | 0.374 | 0.350 | 0.374 | 0.333 | 0.369 | 0.345 | 0.377 | 0.359 | 0.389 | 0.357 | 0.387 | 0.351 | 0.375 | 0.384 | 0.396 | 0.362 | 0.383 |
| | 336 | **0.357** | **0.383** | 0.368 | 0.396 | 0.381 | 0.394 | 0.365 | 0.393 | 0.377 | 0.400 | 0.399 | 0.412 | 0.388 | 0.407 | 0.382 | 0.394 | 0.414 | 0.415 | 0.393 | 0.402 |
| | 720 | **0.412** | **0.415** | 0.426 | 0.426 | 0.435 | 0.424 | 0.421 | 0.424 | 0.432 | 0.433 | 0.463 | 0.450 | 0.448 | 0.443 | 0.436 | 0.428 | 0.501 | 0.463 | 0.447 | 0.434 |
| | Avg | **0.347** | **0.375** | 0.358 | 0.386 | 0.368 | 0.386 | 0.354 | 0.382 | 0.365 | 0.391 | 0.386 | 0.405 | 0.377 | 0.401 | 0.372 | 0.388 | 0.412 | 0.414 | 0.383 | 0.397 |
| ETTm2 | 96 | **0.166** | **0.254** | 0.172 | 0.259 | 0.168 | 0.257 | 0.173 | 0.261 | 0.174 | 0.262 | 0.181 | 0.271 | 0.179 | 0.268 | 0.173 | 0.268 | 0.216 | 0.290 | 0.194 | 0.296 |
| | 192 | **0.225** | **0.290** | 0.233 | 0.298 | 0.234 | 0.301 | 0.237 | 0.307 | 0.239 | 0.305 | 0.244 | 0.313 | 0.239 | 0.309 | 0.248 | 0.328 | 0.262 | 0.323 | 0.262 | 0.343 |
| | 336 | **0.283** | **0.327** | 0.289 | 0.336 | 0.283 | 0.333 | 0.288 | 0.339 | 0.296 | 0.343 | 0.301 | 0.349 | 0.294 | 0.345 | 0.325 | 0.386 | 0.320 | 0.360 | 0.330 | 0.389 |
| | 720 | **0.371** | **0.384** | 0.371 | 0.389 | 0.386 | 0.395 | 0.377 | 0.395 | 0.382 | 0.396 | 0.397 | 0.407 | 0.380 | 0.396 | 0.453 | 0.457 | 0.446 | 0.431 | 0.454 | 0.463 |
| | Avg | **0.261** | **0.314** | 0.267 | 0.321 | 0.268 | 0.321 | 0.269 | 0.326 | 0.273 | 0.327 | 0.281 | 0.335 | 0.273 | 0.330 | 0.300 | 0.360 | 0.311 | 0.351 | 0.310 | 0.373 |
| Rank | | **1.375** | **1.550** | 2.525 | 2.800 | 4.975 | 4.375 | 2.850 | 2.700 | 5.450 | 5.225 | 6.475 | 6.650 | 4.950 | 5.175 | 7.575 | 7.750 | 9.300 | 8.825 | 9.175 | 9.225 |

Final predictions for each channel and patch are then computed via a learnable linear projection:

$$\forall c \in \{1, \ldots, C\}, \ \forall j \in \mathcal{J}_{\text{patch}}, \quad \hat{\mathbf{X}}_{\mathcal{P}_j, c} = \mathbf{Q}_{c,j}^{(1)} \mathbf{W}_{\text{output}} \in \mathbb{R}^P, \quad \text{where } \mathbf{W}_{\text{output}} \in \mathbb{R}^{D \times P}.$$

This query-driven decoding mechanism allows the model to selectively retrieve and reconstruct the missing target patches based on their positional embeddings and contextualized encoder outputs. By actively learning to infer meaningful patterns from diverse and dynamically sampled input contexts, this decoder design enhances the model's ability to capture temporal structure and inter-channel dependencies without relying on explicit supervision. This decoding strategy is a key component of our generalized forecasting formulation.

# 4 Experiments

We design our experiments to investigate the following:

- Does our framework consistently outperform baselines across diverse benchmarks? (§4.1)
- How does each component of our framework contribute to overall performance? (§4.2)
- How is our model designed to organize information flow through attention? (§4.3)

Table 2: Performance comparison across different formulations, encoder attention mechanisms, and positional encoding (PE) sharing strategies. **Bold** indicates the best result within each configuration (*i.e.*, each unique combination of formulation, encoder attention, and PE sharing). A lower MSE or MAE indicates a better performance. The last row in each configuration block corresponds to our proposed model. Each column reports the average performance over four prediction lengths $H \in \{96, 192, 336, 720\}$ with input length $L = 384$.

| Formulation | Encoder Attention | PE Sharing | ETTh1 | | ETTm1 | | Solar | | ECL | |
|---|---|---|---|---|---|---|---|---|---|---|
| | | | MSE | MAE | MSE | MAE | MSE | MAE | MSE | MAE |
| Standard | Latent Bottleneck | Sharing | 0.420 | 0.437 | 0.355 | 0.382 | 0.194 | 0.243 | 0.169 | 0.265 |
| Generalized | Latent Bottleneck | Sharing | **0.404** | **0.420** | **0.338** | **0.370** | **0.182** | **0.233** | **0.157** | **0.249** |
| Generalized | Full Self-Attention | Sharing | 0.425 | 0.434 | 0.353 | 0.375 | 0.192 | 0.237 | 0.161 | 0.254 |
| Generalized | Decoupled Self-Attention | Sharing | 0.423 | 0.433 | 0.356 | 0.379 | 0.189 | **0.233** | 0.158 | 0.250 |
| Generalized | Latent Bottleneck | Sharing | **0.404** | **0.420** | **0.338** | **0.370** | **0.182** | **0.233** | **0.157** | **0.249** |
| Generalized | Latent Bottleneck | Not Sharing | 0.423 | 0.432 | 0.342 | 0.374 | 0.193 | **0.232** | 0.163 | 0.254 |
| Generalized | Latent Bottleneck | Sharing | **0.404** | **0.420** | **0.338** | **0.370** | **0.182** | 0.233 | **0.157** | **0.249** |

**Baselines.** To ensure a comprehensive comparison, we include strong recent baselines such as DeformableTST [7], CARD [8], CATS [9], PatchTST [4], and iTransformer [6], which are Transformer-based, as well as models including SSM-based S-Mamba [17], CNN-based TimesNet [11], and MLP-based DLinear [13], and TiDE [14]. Our focus on Transformer-based methods reflects the architecture of our proposed model. Among these, CARD [8], iTransformer [6], S-Mamba [17], and TimesNet [11] can be categorized as channel-dependent models due to their explicit modeling of inter-variable interactions, while the rest are considered channel-independent. Additionally, we provide comparisons with other baselines in Appendix I.

**Setup.** We evaluate our model on 8 real-world multivariate time series datasets, including ETT (ETTh1, ETTh2, ETTm1 ETTm2) [22], Weather [1], Solar [21], Electricity [2], and Traffic [3] to ensure robustness across diverse temporal patterns and domains. Detailed dataset explanation is described in Appendix A. To ensure fair and robust evaluation with other models, we average results across multiple input lengths {96, 384, 768} for each prediction length {96, 192, 336, 720}, *e.g.*, the main comparisons in Table 1. For results reported in DeformableTST [7], we directly refer to their values, while missing entries are reproduced under the same settings. All other experiments, including ablations and model-specific analysis, are conducted with a fixed input length of 384. Training details can be found in Appendix B.

## 4.1 Multivariate Long-Term Time-Series Forecasting Results

The forecasting performance is summarized in Table 1. We evaluate forecasting performance using MSE and MAE, where lower values indicate better results. Across 8 datasets and 4 forecasting horizons, with two metrics per setting (MSE and MAE), our method achieves 55 best and 17 second best scores out of 80, establishing a new state-of-the-art. Notably, our model significantly outperforms recent channel-dependent methods such as CARD [8] and iTransformer [6]. When averaged across all datasets, our model achieves a 5.6% improvement in MSE and a 4.4% improvement in MAE compared to CARD. Against iTransformer, the improvements are even more substantial, with 8.5% lower MSE and 7.3% lower MAE, highlighting our model's strong capability in capturing complex temporal-channel dependencies.

## 4.2 Component Ablation Studies

We present a component-wise analysis of our forecasting framework, focusing on three key design decisions: the formulation of the forecasting objective, the encoder attention mechanism, and the decoder design. Each choice offers multiple alternatives, and understanding their impact is critical to the final model performance. Table 2 summarizes the performance of various configurations. In the following, we analyze each component in detail and justify the decisions behind our proposed architecture. Full results are provided in Appendix F.

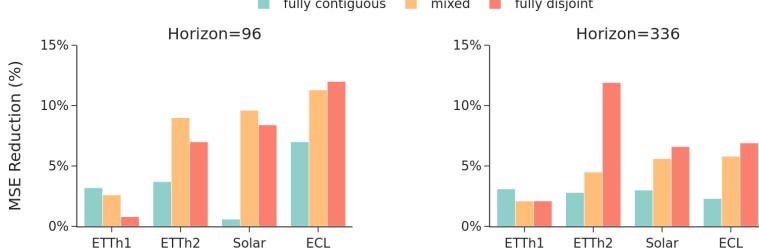

Figure 3: Relative MSE reduction (%) compared to the standard formulation with varying sampling strategies for target patches under input length $L = 384$ and two prediction lengths $H \in \{96, 336\}$. Sampling strategies are (a) fully contiguous: only one contiguous segment in $\mathcal{J}_{\text{patch}}$, (b) fully disjoint: selected target indices are fully arbitrary, and (c) mixed: two equal-sized segments in $\mathcal{J}_{\text{patch}}$. Higher values indicate greater improvements over the standard formulation.

**Formulation.** We compare the *Standard Formulation*, where the model is trained to predict future values given a continuous segment of past observations, against the *Generalized Formulation*, which unifies extrapolation, interpolation, and imputation within a forecasting task described in Section 3.1. This broader formulation allows the model to learn from more diverse temporal prediction objectives by randomly sampling prediction targets along the temporal axis. The results show that our generalized formulation consistently yields better performance across all datasets and metrics, achieving an average improvement of 5.0% in MSE and 3.4% in MAE. This demonstrates the benefit of exposing the model to a wider range of temporal reasoning tasks during training, improving its ability to generalize across different forecasting scenarios.

Our generalized formulation can be easily applied to other patch-based architectures by replacing their prediction layers with our query-based decoder. PatchTST [4] is a clear example: replacing its linear decoder by our query-based decoder and adopting it with our generalized formulation yields a TIMEPERCEIVER variant that uses PatchTST's encoder design. As shown in Table 3, we observe that training with the generalized formulation consistently outperforms the standard formulation. This result indicates that our generalized formulation is broadly applicable across different model designs.

Table 3: Performance comparison of the generalized formulation with PatchTST [4] on the ETTh1 dataset.

| Model | TimePerceiver | | PatchTST | |
|---|---|---|---|---|
| Metric | MSE | MAE | MSE | MAE |
| Standard | 0.420 | 0.437 | 0.423 | 0.436 |
| Generalized | **0.404** | **0.420** | **0.415** | **0.431** |

To further analyze our generalized formulation, we implement three different patch sampling strategies: (a) selecting target patch indices $\mathcal{J}_{\text{patch}}$ contiguously (*i.e.*, fully contiguous), (b) selecting $\mathcal{J}_{\text{patch}}$ randomly (*i.e.*, fully disjoint), and (c) selecting $\mathcal{J}_{\text{patch}}$ including two contiguous, equal-sized segments (*i.e.*, mixed). The forecasting results are reported in Figure 3. All the strategies contribute to performance improvement, but the optimal configuration varies with dataset characteristics and prediction lengths. For example, while ETTh1 favors contiguous sampling, others benefit more from disjoint or mixed strategies, even among similar datasets such as ETTh1 and ETTh2. This highlights the importance of selecting an appropriate sampling ratio based on the specific forecasting context. Without hyperparameter tuning, selecting two equal-sized segments consistently yields reliable performance and serves as a robust default.

**Encoder Attention.** We compare different encoder attention mechanisms to understand how temporal and channel dependencies should be modeled. The *Self-Attention* approach directly applies attention across input tokens, either jointly across temporal and channel dimensions or by splitting them. While simple, this method incurs a large memory and computation cost, especially as the number of input channels or time steps grows. Moreover, attending over all input tokens jointly can lead to diluted attention scores and unstable optimization. To address these issues, we adopt the *Latent Bottleneck* approach, where a fixed set of latent tokens attends to the input. This design enables the model to compress relevant information from the input space into a smaller latent space, reducing the attention complexity from $\mathcal{O}(N^2)$ to $\mathcal{O}(NM)$, where $M \ll N$ is the number of latent tokens, as further analyzed in Appendix D. Unlike axis-separated attention, this latent-based method captures temporal and inter-channel dependencies jointly, leading to more expressive and efficient representations. Our results show that *Latent Bottleneck* outperforms both self-attention variants across all datasets.

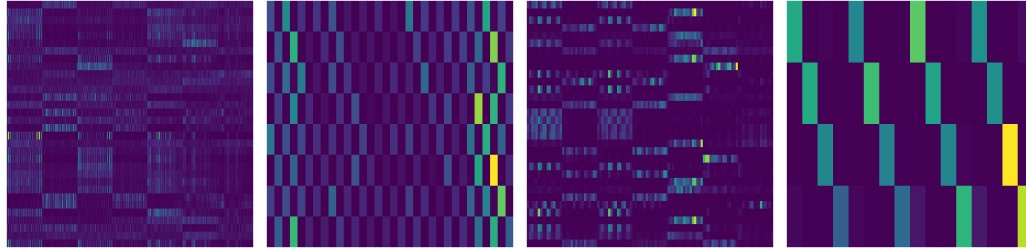

(a) Encoder and decoder attention on **ETTh1**   (b) Encoder and decoder attention on **ETTm1**

Figure 4: Cross-attention maps from the encoder and decoder on ETTh1 and ETTm1. For each dataset, the left image shows attention from latent tokens to input patches (encoder), and the right shows attention from query tokens to input patches (decoder). In encoder maps, the x-axis shows flattened input patch indices ($C \times |\mathcal{I}_{\text{patch}}|$), and the y-axis represents latent tokens ($M$). In decoder maps, the x-axis is temporal input positions ($|\mathcal{I}_{\text{patch}}|$, $L{=}384$), and the y-axis is query tokens aligned with prediction patches ($|\mathcal{J}_{\text{patch}}|$, $H{=}96$). Each token spans 12 hours (ETTh1) or 6 hours (ETTm1).

**Decoder Design.** In query-based decoding, it is essential for the model to distinguish between time points used as input and those expected as output. To enable this, we apply both temporal and channel positional embeddings, TPE and CPE, not only into the encoder but also into the decoder queries. We compare two strategies: one where the encoder and decoder share the same positional embeddings, and another where they use separate embeddings. As shown in Table 2, using shared positional embeddings generally leads to better performance across most datasets, supporting the importance of temporal alignment between encoder inputs and decoder queries. An exception is observed in the Solar dataset, where non-shared embeddings slightly outperform the shared version in terms of MAE.

## 4.3 Information Flow Analysis

**Use of Latents.** We conduct experiments with a variant, denoted as DirectLT, in which the decoder directly consumes latent tokens without re-expansion to full resolution. As demonstrated in Table 4, our decoder design yields substantially better performance compared to directly employing the encoded latent tokens as decoder inputs. The superiority of our design can be attributed to the following reasoning. In our framework, the latent bottleneck is designed to capture global temporal-channel patterns, while the original input sequence retains fine-grained local signals. Relying solely on the latent makes it difficult to reconstruct these detailed signals, ultimately leading to performance degradation. In contrast, our framework enriches input patch representations by leveraging the latent as an auxiliary memory, enabling each to incorporate both global contexts and local details.

Table 4: Performance comparison where the decoder directly uses latents (referred to as DirectLT) on ETTh1.

| Model | Ours | | DirectLT | |
|---|---|---|---|---|
| Metric | MSE | MAE | MSE | MAE |
| 96 | **0.366** | **0.393** | 0.402 | 0.418 |
| 192 | **0.394** | **0.411** | 0.426 | 0.437 |
| 336 | **0.413** | **0.422** | 0.429 | 0.438 |
| 720 | **0.445** | **0.453** | 0.473 | 0.472 |
| Avg | **0.404** | **0.420** | 0.433 | 0.441 |

**Cross Attention.** To better understand how our model processes information throughout encoding and decoding, we visualize the cross-attention maps in Figure 4, using examples from the ETTh1 and ETTm1 datasets. For each dataset, the left side of the figure shows encoder-side attention from latent tokens to input patches, while the right side depicts decoder-side attention from query tokens to input patches. The encoder attention maps reveal that latent tokens distribute their focus across diverse regions of the input, spanning both temporal and channel dimensions. Rather than converging on a single dominant region, each latent appears to capture distinct patterns or features, indicating that the model effectively compresses input information while maintaining representational diversity.

On the decoder side, the attention maps illustrate how query tokens attend to input patches in a temporally structured manner. Queries often align with regularly spaced regions in the input, suggesting that the model learns to exploit periodic patterns in the data. In ETTh1, where each input patch covers 12 hours, decoder queries commonly attend to patches spaced about two positions apart, which roughly corresponds to a daily cycle (24 hours). In ETTm1, where each patch covers a fixed time span of 6 hours, the attention aligns with intervals of four positions, reflecting a quarter-day

rhythm. These resolution-aware patterns demonstrate the model's ability to adapt its decoding strategy to the temporal resolution of the dataset and capture underlying periodic structure effectively.

## 5 Related Work

Recent time-series forecasting research has mainly focused on encoder design, particularly on modeling channel interactions, categorized as channel-independent (CI) [4, 7, 9, 13, 14] or channel-dependent (CD) approaches [5, 6, 8, 11, 17]. While CD methods leverage richer inter-channel information, effectively utilizing it remains challenging, leading to recent attempts to capture both temporal and cross-channel dependencies separately [5, 6]. Meanwhile, most models still rely on simple linear projections [4, 6, 7] from encoder outputs for prediction, using a uniform training formulation [6–11, 13, 14, 17]. Although query-based decoders [9] have recently developed for time-series forecasting, they do not study how to effectively encode the input time series. In contrast, our unified framework leverages a generalized formulation with an optimized structure.

## 6 Conclusion

In this work, we propose TimePerceiver, a unified framework that integrates architectural design and training strategy for multivariate time-series forecasting. Specifically, we introduce a generalized forecasting formulation that encompasses extrapolation, interpolation, and imputation. To support this formulation, we design a novel encoder-decoder architecture equipped with latent bottleneck representations and query-based decoding, enabling flexible modeling of arbitrarily positioned input and target segments. Extensive experiments across diverse benchmarks demonstrate that our framework consistently outperforms prior state-of-the-art methods. We hope our work encourages the community to explore holistic modeling approaches that align architectural choices more closely with the underlying forecasting objectives.

**Limitations and Broader Impacts.** Please refer to Appendix J for further discussion.

## Acknowledgment

This work was partly supported by the grants from Institute of Information & Communications Technology Planning & Evaluation (IITP), funded by the Korean government (MSIT; Ministry of Science and ICT): No. RS-2019-II190421, No. IITP-2025-RS-2020-II201821, No. IITP-2025-RS-2024-00437633, and No. RS-2025-25442569.

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

# Appendix

## A  Dataset

Table 5: Details of benchmark datasets used in our experiments.

| Dataset | Variables | Frequency | Time Steps | Domain |
|---------|-----------|-----------|------------|--------|
| ETTh1 | 7 | 1 hour | 17,420 | Temperature |
| ETTh2 | 7 | 1 hour | 17,420 | Temperature |
| ETTm1 | 7 | 15 minutes | 69,680 | Temperature |
| ETTm2 | 7 | 15 minutes | 69,680 | Temperature |
| Weather | 21 | 10 minutes | 52,696 | Weather |
| Solar | 137 | 10 minutes | 52,560 | Energy |
| Electricity | 321 | 1 hour | 26,304 | Electricity |
| Traffic | 862 | 1 hour | 17,544 | Transportation |

We utilize 8 widely adopted benchmark datasets for time series forecasting: ETT (ETTh1, ETTh2, ETTm1, ETTm2) [22], Weather [1], Solar [21], Electricity [2], and Traffic [3]. These datasets are commonly used in recent time series forecasting literature and span a wide range of temporal resolutions, domains, and sequence characteristics. Descriptions of each dataset are provided as follows, while detailed statistics such as data size and sampling frequency are summarized in Table 5.

- **ETT** [22] contains 7 variables from two electricity transformers between 2016 and 2018, with four subsets: ETTh1/ETTh2 (hourly) and ETTm1/ETTm2 (every 15 minutes).
- **Weather** [1] contains 21 meteorological variables collected from a weather station in Germany during 2020.
- **Solar** [21] contains 137 variables representing solar power generation from photovoltaic plants in Alabama during 2006.
- **Electricity** [2] contains hourly electricity consumption variables from 321 clients, collected between 2012 and 2014.
- **Traffic** [3] contains road occupancy rate variables measured hourly by 862 sensors on San Francisco Bay Area freeways, covering the period from 2015 to 2016.

Following the standard protocol [7, 22] widely adopted in time series forecasting research, we perform a chronological split of each dataset. For the ETT datasets, we use a 6:2:2 ratio for training, validation, and testing. For the remaining datasets—Weather, Solar, Electricity, and Traffic—we apply a 7:1:2 split.

# B Implementation Details

---

**Algorithm 1** TIMEPERCEIVER

---

**Require:** Concatenated input token sequence $\mathbf{X}_{\mathcal{I}_{\text{patch}}} \in \mathbb{R}^{C \times (|\mathcal{I}_{\text{patch}}|P)}$, input token indices $\mathcal{I}_{\text{patch}} \subset \{1, 2, \ldots, N\}$, target token indices $\mathcal{J}_{\text{patch}} = \{1, \ldots, N\} \setminus \mathcal{I}_{\text{patch}}$, patch length $P$, embedding dimension $D$, number of latent tokens $M$, latent dimension $D_L$

1: Apply instance normalization (RevIN) to $\mathbf{X}_{\mathcal{I}_{\text{patch}}}$ and store $(\boldsymbol{\mu}, \boldsymbol{\sigma})$
2: $\mathbf{H}^{(0)} = \mathbf{X}_{\mathcal{I}_{\text{patch}}}\mathbf{W}_{\text{input}} + \mathbf{E}^{\text{channel}} + \mathbf{E}^{\text{temporal}}$      $\triangleright \mathbf{H}^{(0)} \in \mathbb{R}^{(C|\mathcal{I}_{\text{patch}}|) \times D}$
    // Encode input into latent tokens (information compression)
3: Use learnable latent array $\mathbf{Z}^{(0)}$      $\triangleright \mathbf{Z}^{(0)} \in \mathbb{R}^{M \times D_L}$
4: $\mathbf{Z}^{(1)} = \texttt{AttnBlock}(\mathbf{Z}^{(0)}, \mathbf{H}^{(0)}, \mathbf{H}^{(0)})$      $\triangleright \mathbf{Z}^{(1)} \in \mathbb{R}^{M \times D_L}$
5: **for** $k = 1$ to $K$ **do**
6:      $\mathbf{Z}^{(k+1)} = \texttt{AttnBlock}(\mathbf{Z}^{(k)}, \mathbf{Z}^{(k)}, \mathbf{Z}^{(k)})$
7: **end for**
    // Update input tokens using refined latent representation
8: $\mathbf{H}^{(1)} = \texttt{AttnBlock}(\mathbf{H}^{(0)}, \mathbf{Z}^{(K+1)}, \mathbf{Z}^{(K+1)})$      $\triangleright \mathbf{H}^{(1)} \in \mathbb{R}^{(C|\mathcal{I}_{\text{patch}}|) \times D}$
    // Predict target tokens by attending to updated input representation
9: Use learnable query $\mathbf{Q}^{(0)}$      $\triangleright \mathbf{Q}^{(0)} \in \mathbb{R}^{(C|\mathcal{J}_{\text{patch}}|) \times D}$
10: $\mathbf{Q}^{(1)} = \texttt{AttnBlock}(\mathbf{Q}^{(0)}, \mathbf{H}^{(1)}, \mathbf{H}^{(1)})$      $\triangleright \mathbf{Q}^{(1)} \in \mathbb{R}^{(C|\mathcal{J}_{\text{patch}}|) \times D}$
    // Project query to patch-level predictions
11: $\hat{\mathbf{X}}_{\mathcal{J}_{\text{patch}}} = \mathbf{Q}^{(1)}\mathbf{W}_{\text{output}}$      $\triangleright \hat{\mathbf{X}}_{\mathcal{J}_{\text{patch}}} \in \mathbb{R}^{C \times |\mathcal{J}_{\text{patch}}|P}$
12: Apply instance denormalization (RevIN) to restore original scale

---

**Hyperparameter Settings.** For the TIMEPERCEIVER parameter $\theta$, we use the AdamW optimizer [26] with a weight decay of $0.05$. The learning rate is set to $1 \times 10^{-4}$ for the Weather dataset and $5 \times 10^{-4}$ for all other datasets, with a warmup phase of 5 epochs. We train for 50 epochs on the ETT and Solar datasets, and 100 epochs on all others. The batch size is set to 32 for the Solar, Electricity, and Traffic datasets, and 128 for the remaining datasets. We choose the patch size $P \in \{12, 16, 24, 48\}$ and the embedding dimension $D \in \{256, 512\}$. The number of latents is set to $M \in \{8, 16, 32\}$, and the latent dimension to $D_L \in \{64, 128, 256\}$. For simplicity, we denote all latent dimensions as $D$ in section 3.3, although in practice we use different values for different components. For the attention modules, we choose the number of heads $n_{\text{heads}} \in \{4, 8\}$ and feedforward dimension $d_{\text{ff}}$ in each attention module is set to twice the query dimension. Lastly, the separate ratio is selected from $\{0, 0.5, 1\}$, corresponding to the fully disjoint, mixed, and fully contiguous sampling settings, respectively, as illustrated in Figure 3. All results are reported as the average over 5 runs with different random seeds.

**Instance Normalization.** To address distribution shifts between training and test data, we apply Reversible Instance Normalization (RevIN) [27]. Each input is normalized before processing, and the original statistics are restored after prediction.

**Reproducibility Statement.** We provide our code with training and evaluation scripts to ensure reproducibility at https://github.com/efficient-learning-lab/TimePerceiver. All the experiments were conducted using NVIDIA RTX 4090 GPUs with the PyTorch [28].

# C  Error Bar

Table 6: Standard deviation of performance results with input length $L = 384$ and prediction lengths $H \in \{96, 192, 336, 720\}$ on five random seeds.

| Dataset | Weather | | Solar | | ECL | | Traffic | |
|---|---|---|---|---|---|---|---|---|
| Metric | MSE | MAE | MSE | MAE | MSE | MAE | MSE | MAE |
| 96 | 0.144±0.002 | 0.195±0.003 | 0.163±0.006 | 0.213±0.005 | 0.125±0.001 | 0.219±0.003 | 0.373±0.001 | 0.262±0.002 |
| 192 | 0.190±0.003 | 0.243±0.001 | 0.176±0.003 | 0.228±0.002 | 0.142±0.002 | 0.235±0.001 | 0.383±0.002 | 0.263±0.003 |
| 336 | 0.242±0.000 | 0.279±0.001 | 0.185±0.003 | 0.246±0.004 | 0.161±0.003 | 0.254±0.004 | 0.399±0.002 | 0.270±0.003 |
| 720 | 0.315±0.003 | 0.336±0.001 | 0.197±0.005 | 0.244±0.003 | 0.200±0.002 | 0.288±0.003 | 0.433±0.003 | 0.289±0.002 |

| Dataset | ETTh1 | | ETTh2 | | ETTm1 | | ETTm2 | |
|---|---|---|---|---|---|---|---|---|
| Metric | MSE | MAE | MSE | MAE | MSE | MAE | MSE | MAE |
| 96 | 0.366±0.004 | 0.393±0.003 | 0.272±0.004 | 0.333±0.005 | 0.283±0.007 | 0.334±0.004 | 0.161±0.005 | 0.257±0.004 |
| 192 | 0.394±0.002 | 0.411±0.002 | 0.333±0.003 | 0.381±0.004 | 0.321±0.004 | 0.359±0.002 | 0.215±0.003 | 0.285±0.004 |
| 336 | 0.413±0.006 | 0.422±0.006 | 0.346±0.009 | 0.399±0.011 | 0.352±0.004 | 0.378±0.002 | 0.273±0.003 | 0.325±0.004 |
| 720 | 0.445±0.008 | 0.453±0.004 | 0.386±0.010 | 0.431±0.009 | 0.399±0.002 | 0.410±0.002 | 0.357±0.006 | 0.375±0.005 |

We evaluate the stability of TIMEPERCEIVER by reporting mean and standard deviation over five runs with lookback window $L = 384$. As shown in Table 6, across all 64 experimental settings, 54 cases exhibit a standard deviation less than or equal to 0.0005, indicating that TIMEPERCEIVER remains highly stable.

# D Computation Efficiency

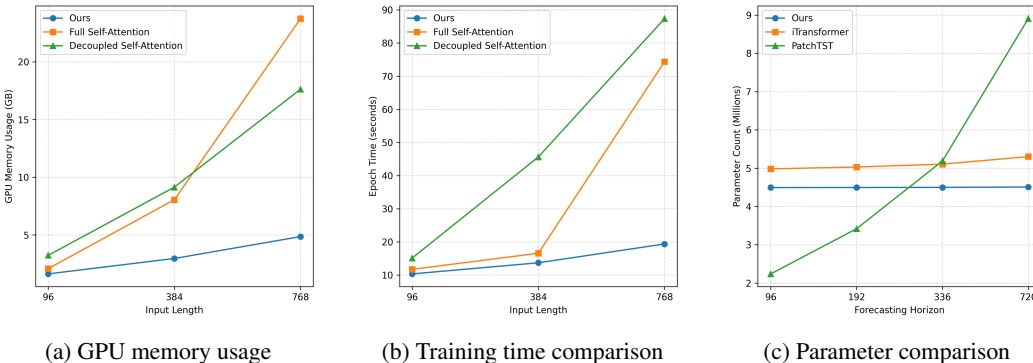

(a) GPU memory usage      (b) Training time comparison      (c) Parameter comparison

Figure 5: (a) and (b) compare computational costs resulting from different encoder structures, including GPU memory usage and epoch time per input length. (c) shows the difference in parameter count between our model and other commonly used models with decoder structures based on linear projection, across various forecasting horizons.

Table 7: Computation cost comparison with other baselines.

| Dataset | TimePerceiver | | DeformableTST | | CARD | | PatchTST | | Crossformer | | TimesNet | |
|---|---|---|---|---|---|---|---|---|---|---|---|---|
| Input length | 96 | 384 | 96 | 384 | 96 | 384 | 96 | 384 | 96 | 384 | 96 | 384 |
| Memory(MB) | 660 | 918 | 972 | 2998 | 538 | 724 | 1512 | 6852 | 2768 | 4110 | 1300 | 3258 |
| Runtime(iter/s) | 0.0435 | 0.0583 | 0.0919 | 0.1153 | 0.0737 | 0.0828 | 0.0547 | 0.1156 | 0.1299 | 0.1350 | 0.1376 | 0.1385 |

We empirically validate the efficiency of our proposed encoder design through comparisons across three variants: our model with a latent bottleneck, full self-attention, and decoupled self-attention. As shown in Figure 5a and Figure 5b, our method consistently achieves lower GPU memory usage and faster training time, particularly as the input length increases. This is attributed to the use of a small set of latent tokens, which enables the model to abstract and interact with the input more efficiently than direct token-to-token computation. In contrast, full self-attention suffers from quadratic complexity with respect to input length. Decoupled self-attention, which applies attention separately along the temporal and channel dimensions, avoids this quadratic scaling but still incurs higher overhead due to its dual attention structure.

Furthermore, Figure 5c highlights the parameter efficiency of query-based decoder [9]. Since each query is projected through a shared linear layer, our model maintains a nearly constant number of parameters regardless of the forecasting horizon. In contrast, models employing direct linear projection from encoded representations [4, 6], exhibit a linear growth in parameter count as the prediction length increases. These results show that our design remains efficient when handling both long input sequences and extended forecasting horizons.

Beyond these aspects, Table 7 shows that TIMEPERCEIVER demonstrates efficiency in both training time and memory footprint compared to baselines [4, 5, 7, 8, 11], regardless of the input length. This efficiency stems from the latent bottleneck that enables compression of core information efficiently.

# E  Full Results of Multivariate Long-Term Forecasting

To complement the main results reported in Table 1, we report the full forecasting performance for each input length $L \in \{96, 384, 768\}$ in this appendix. These detailed results provide a more comprehensive view of model behavior across varying lookback window size.

Table 8: Multivariate long-term forecasting results compared with baseline models. **Bold** indicates the best result, and underlined denotes the second best. The results are reported using input length $L = 96$ and prediction lengths $H \in \{96, 192, 336, 720\}$.

| Models | | **TimePerceiver** | | DeformableTST | | CARD | | CATS | | PatchTST | | iTransformer | | S-Mamba | | DLinear | | TimesNet | | TiDE | |
|---|---|---|---|---|---|---|---|---|---|---|---|---|---|---|---|---|---|---|---|---|---|
| Metric | | MSE | MAE | MSE | MAE | MSE | MAE | MSE | MAE | MSE | MAE | MSE | MAE | MSE | MAE | MSE | MAE | MSE | MAE | MSE | MAE |
| Weather | 96 | **0.153** | **0.202** | 0.169 | 0.213 | 0.160 | 0.208 | 0.161 | 0.207 | 0.177 | 0.218 | 0.174 | 0.214 | 0.165 | 0.210 | 0.196 | 0.255 | 0.172 | 0.220 | 0.202 | 0.261 |
| | 192 | **0.202** | **0.247** | 0.216 | 0.255 | 0.207 | 0.250 | 0.208 | 0.250 | 0.225 | 0.259 | 0.221 | 0.254 | 0.214 | 0.252 | 0.237 | 0.296 | 0.219 | 0.261 | 0.242 | 0.298 |
| | 336 | **0.258** | **0.288** | 0.271 | 0.294 | 0.265 | 0.292 | 0.264 | 0.290 | 0.278 | 0.297 | 0.278 | 0.296 | 0.274 | 0.297 | 0.283 | 0.335 | 0.280 | 0.306 | 0.287 | 0.335 |
| | 720 | **0.338** | **0.340** | 0.347 | 0.344 | 0.346 | 0.346 | 0.342 | 0.341 | 0.354 | 0.348 | 0.358 | 0.347 | 0.350 | 0.345 | 0.345 | 0.381 | 0.365 | 0.359 | 0.351 | 0.386 |
| | Avg | **0.238** | **0.269** | 0.251 | 0.276 | 0.245 | 0.274 | 0.244 | 0.272 | 0.259 | 0.281 | 0.258 | 0.278 | 0.251 | 0.276 | 0.265 | 0.317 | 0.259 | 0.287 | 0.271 | 0.320 |
| Solar | 96 | 0.199 | 0.238 | **0.193** | **0.232** | 0.230 | 0.281 | 0.216 | 0.247 | 0.234 | 0.286 | 0.203 | 0.237 | 0.205 | 0.244 | 0.290 | 0.378 | 0.250 | 0.292 | 0.312 | 0.399 |
| | 192 | 0.228 | 0.259 | **0.225** | 0.261 | 0.267 | 0.308 | 0.247 | 0.267 | 0.267 | 0.310 | 0.233 | 0.261 | 0.237 | 0.270 | 0.320 | 0.398 | 0.296 | 0.318 | 0.339 | 0.416 |
| | 336 | 0.247 | 0.273 | **0.239** | **0.267** | 0.289 | 0.319 | 0.267 | 0.277 | 0.290 | 0.315 | 0.248 | 0.273 | 0.258 | 0.288 | 0.353 | 0.415 | 0.319 | 0.330 | 0.368 | 0.430 |
| | 720 | 0.257 | 0.281 | **0.238** | **0.265** | 0.294 | 0.327 | 0.296 | 0.293 | 0.289 | 0.317 | 0.249 | 0.275 | 0.260 | 0.288 | 0.356 | 0.413 | 0.338 | 0.337 | 0.370 | 0.425 |
| | Avg | 0.233 | 0.263 | **0.224** | **0.256** | 0.270 | 0.309 | 0.257 | 0.271 | 0.270 | 0.307 | 0.233 | 0.262 | 0.240 | 0.273 | 0.330 | 0.401 | 0.301 | 0.319 | 0.347 | 0.417 |
| Electricity | 96 | **0.131** | **0.225** | 0.158 | 0.255 | 0.155 | 0.246 | 0.149 | 0.237 | 0.181 | 0.270 | 0.148 | 0.240 | 0.139 | 0.235 | 0.197 | 0.282 | 0.168 | 0.272 | 0.237 | 0.329 |
| | 192 | **0.150** | **0.244** | 0.168 | 0.265 | 0.170 | 0.259 | 0.163 | 0.250 | 0.188 | 0.274 | 0.162 | 0.253 | 0.159 | 0.255 | 0.196 | 0.285 | 0.184 | 0.289 | 0.236 | 0.330 |
| | 336 | **0.168** | **0.263** | 0.183 | 0.280 | 0.194 | 0.285 | 0.180 | 0.268 | 0.204 | 0.293 | 0.178 | 0.269 | 0.176 | 0.272 | 0.209 | 0.301 | 0.198 | 0.300 | 0.249 | 0.344 |
| | 720 | 0.209 | **0.298** | 0.223 | 0.313 | 0.229 | 0.316 | 0.219 | 0.302 | 0.246 | 0.324 | 0.225 | 0.317 | **0.204** | **0.298** | 0.245 | 0.333 | 0.220 | 0.320 | 0.284 | 0.373 |
| | Avg | **0.165** | **0.258** | 0.183 | 0.278 | 0.187 | 0.277 | 0.178 | 0.264 | 0.205 | 0.290 | 0.178 | 0.270 | 0.170 | 0.265 | 0.212 | 0.300 | 0.192 | 0.295 | 0.251 | 0.344 |
| Traffic | 96 | 0.400 | 0.263 | 0.418 | 0.271 | 0.448 | 0.300 | 0.421 | 0.270 | 0.462 | 0.295 | 0.395 | 0.268 | **0.382** | **0.261** | 0.650 | 0.396 | 0.593 | 0.321 | 0.805 | 0.493 |
| | 192 | 0.416 | **0.265** | 0.437 | 0.276 | 0.465 | 0.303 | 0.436 | 0.275 | 0.466 | 0.296 | 0.417 | 0.276 | **0.396** | 0.267 | 0.598 | 0.370 | 0.617 | 0.336 | 0.756 | 0.474 |
| | 336 | 0.430 | 0.276 | 0.449 | 0.290 | 0.477 | 0.306 | 0.453 | 0.284 | 0.482 | 0.304 | 0.433 | 0.283 | **0.417** | **0.276** | 0.605 | 0.373 | 0.629 | 0.336 | 0.762 | 0.477 |
| | 720 | 0.467 | **0.297** | 0.477 | 0.303 | 0.509 | 0.323 | 0.484 | 0.303 | 0.514 | 0.322 | 0.467 | 0.302 | **0.460** | 0.300 | 0.645 | 0.394 | 0.640 | 0.350 | 0.719 | 0.449 |
| | Avg | 0.428 | 0.275 | 0.445 | 0.285 | 0.475 | 0.308 | 0.450 | 0.283 | 0.481 | 0.304 | 0.428 | 0.282 | **0.414** | **0.276** | 0.625 | 0.383 | 0.620 | 0.336 | 0.760 | 0.473 |
| ETTh1 | 96 | 0.372 | **0.394** | 0.373 | 0.396 | 0.398 | 0.405 | **0.371** | 0.395 | 0.414 | 0.419 | 0.386 | 0.405 | 0.386 | 0.405 | 0.386 | 0.400 | 0.384 | 0.402 | 0.479 | 0.464 |
| | 192 | **0.422** | 0.425 | 0.427 | 0.427 | 0.448 | 0.433 | 0.426 | **0.422** | 0.460 | 0.445 | 0.441 | 0.436 | 0.443 | 0.437 | 0.437 | 0.432 | 0.436 | 0.429 | 0.525 | 0.492 |
| | 336 | 0.444 | 0.434 | **0.437** | **0.426** | 0.494 | 0.457 | **0.437** | 0.432 | 0.501 | 0.466 | 0.487 | 0.458 | 0.489 | 0.468 | 0.481 | 0.459 | 0.491 | 0.469 | 0.565 | 0.515 |
| | 720 | **0.451** | **0.453** | 0.464 | 0.462 | 0.483 | 0.472 | 0.474 | 0.461 | 0.500 | 0.488 | 0.503 | 0.491 | 0.502 | 0.489 | 0.519 | 0.516 | 0.521 | 0.500 | 0.594 | 0.558 |
| | Avg | **0.422** | **0.427** | 0.425 | 0.428 | 0.456 | 0.442 | 0.427 | 0.428 | 0.469 | 0.454 | 0.454 | 0.447 | 0.455 | 0.450 | 0.456 | 0.452 | 0.458 | 0.450 | 0.541 | 0.507 |
| ETTh2 | 96 | 0.285 | 0.338 | **0.281** | **0.334** | 0.294 | 0.341 | 0.287 | 0.341 | 0.302 | 0.348 | 0.297 | 0.349 | 0.296 | 0.348 | 0.333 | 0.387 | 0.340 | 0.374 | 0.400 | 0.440 |
| | 192 | 0.365 | 0.388 | **0.353** | **0.382** | 0.375 | 0.391 | 0.361 | 0.388 | 0.388 | 0.400 | 0.380 | 0.400 | 0.376 | 0.396 | 0.477 | 0.476 | 0.402 | 0.414 | 0.528 | 0.509 |
| | 336 | 0.366 | 0.414 | **0.341** | **0.379** | 0.419 | 0.426 | 0.374 | 0.403 | 0.426 | 0.433 | 0.428 | 0.432 | 0.424 | 0.431 | 0.594 | 0.541 | 0.452 | 0.452 | 0.643 | 0.571 |
| | 720 | **0.404** | 0.440 | 0.410 | **0.431** | 0.420 | 0.438 | 0.412 | 0.433 | 0.431 | 0.446 | 0.427 | 0.445 | 0.426 | 0.444 | 0.831 | 0.657 | 0.462 | 0.468 | 0.874 | 0.679 |
| | Avg | 0.355 | 0.395 | **0.346** | **0.382** | 0.378 | 0.399 | 0.359 | 0.391 | 0.387 | 0.407 | 0.383 | 0.407 | 0.381 | 0.405 | 0.559 | 0.515 | 0.414 | 0.427 | 0.611 | 0.550 |
| ETTm1 | 96 | **0.306** | **0.347** | 0.316 | 0.358 | 0.327 | 0.359 | 0.318 | **0.347** | 0.329 | 0.367 | 0.334 | 0.368 | 0.333 | 0.368 | 0.345 | 0.372 | 0.338 | 0.375 | 0.364 | 0.387 |
| | 192 | **0.343** | **0.369** | 0.354 | 0.380 | 0.372 | 0.381 | 0.357 | 0.377 | 0.367 | 0.385 | 0.377 | 0.391 | 0.376 | 0.390 | 0.380 | 0.389 | 0.374 | 0.387 | 0.398 | 0.404 |
| | 336 | **0.371** | **0.391** | 0.379 | 0.405 | 0.403 | 0.401 | 0.387 | 0.410 | 0.399 | 0.410 | 0.426 | 0.420 | 0.408 | 0.413 | 0.413 | 0.413 | 0.410 | 0.411 | 0.428 | 0.425 |
| | 720 | **0.428** | **0.426** | 0.443 | 0.438 | 0.467 | 0.438 | 0.448 | 0.437 | 0.454 | 0.439 | 0.491 | 0.459 | 0.475 | 0.448 | 0.474 | 0.453 | 0.478 | 0.450 | 0.487 | 0.461 |
| | Avg | **0.362** | **0.383** | 0.373 | 0.395 | 0.392 | 0.395 | 0.378 | 0.393 | 0.387 | 0.400 | 0.407 | 0.410 | 0.398 | 0.405 | 0.403 | 0.407 | 0.400 | 0.406 | 0.419 | 0.419 |
| ETTm2 | 96 | **0.170** | **0.249** | 0.178 | 0.262 | 0.176 | 0.259 | 0.178 | 0.261 | 0.175 | 0.259 | 0.180 | 0.264 | 0.179 | 0.263 | 0.193 | 0.292 | 0.187 | 0.267 | 0.207 | 0.305 |
| | 192 | **0.236** | **0.293** | 0.243 | 0.301 | 0.246 | 0.306 | 0.248 | 0.308 | 0.241 | 0.302 | 0.250 | 0.309 | 0.250 | 0.309 | 0.284 | 0.362 | 0.249 | 0.309 | 0.290 | 0.364 |
| | 336 | **0.298** | **0.330** | 0.310 | 0.348 | 0.302 | 0.343 | 0.304 | 0.343 | 0.305 | 0.343 | 0.311 | 0.348 | 0.312 | 0.349 | 0.369 | 0.427 | 0.321 | 0.351 | 0.377 | 0.422 |
| | 720 | **0.399** | **0.392** | 0.400 | 0.398 | 0.400 | 0.404 | 0.402 | 0.402 | 0.402 | 0.400 | 0.412 | 0.407 | 0.411 | 0.406 | 0.554 | 0.522 | 0.408 | 0.403 | 0.558 | 0.524 |
| | Avg | **0.276** | **0.316** | 0.283 | 0.327 | 0.281 | 0.328 | 0.283 | 0.329 | 0.281 | 0.326 | 0.288 | 0.332 | 0.288 | 0.332 | 0.350 | 0.401 | 0.291 | 0.333 | 0.358 | 0.404 |

Table 9: Multivariate long-term forecasting results compared with baseline models. **Bold** indicates the best result, and underlined denotes the second best. The results are reported using input length $L = 384$ and prediction lengths $H \in \{96, 192, 336, 720\}$.

| Models | | **TimePerceiver** | | DeformableTST | | CARD | | CATS | | PatchTST | | iTransformer | | S-Mamba | | DLinear | | TimesNet | | TiDE | |
|---|---|---|---|---|---|---|---|---|---|---|---|---|---|---|---|---|---|---|---|---|---|
| Metric | | MSE | MAE | MSE | MAE | MSE | MAE | MSE | MAE | MSE | MAE | MSE | MAE | MSE | MAE | MSE | MAE | MSE | MAE | MSE | MAE |
| Weather | 96 | **0.144** | **0.195** | 0.149 | 0.198 | 0.155 | 0.209 | 0.147 | 0.199 | 0.150 | 0.201 | 0.160 | 0.210 | 0.157 | 0.208 | 0.172 | 0.232 | 0.161 | 0.216 | 0.176 | 0.228 |
| | 192 | **0.190** | 0.243 | 0.192 | **0.238** | 0.207 | 0.255 | 0.191 | 0.242 | 0.194 | 0.244 | 0.203 | 0.251 | 0.203 | 0.250 | 0.215 | 0.272 | 0.234 | 0.278 | 0.219 | 0.263 |
| | 336 | **0.242** | 0.279 | 0.244 | **0.278** | 0.263 | 0.294 | 0.244 | 0.280 | 0.243 | 0.279 | 0.252 | 0.287 | 0.256 | 0.290 | 0.264 | 0.317 | 0.286 | 0.316 | 0.266 | 0.298 |
| | 720 | 0.315 | 0.336 | 0.317 | **0.331** | 0.355 | 0.352 | **0.314** | **0.331** | 0.319 | 0.333 | 0.325 | 0.339 | 0.322 | 0.336 | 0.324 | 0.364 | 0.373 | 0.368 | 0.333 | 0.345 |
| | Avg | **0.223** | 0.263 | 0.225 | **0.261** | 0.245 | 0.277 | 0.224 | 0.263 | 0.226 | 0.264 | 0.235 | 0.272 | 0.235 | 0.272 | 0.244 | 0.296 | 0.264 | 0.295 | 0.249 | 0.284 |
| Solar | 96 | **0.163** | **0.213** | 0.173 | 0.241 | 0.175 | 0.243 | 0.172 | 0.216 | 0.197 | 0.291 | 0.187 | 0.246 | 0.194 | 0.243 | 0.216 | 0.316 | 0.217 | 0.311 | 0.228 | 0.300 |
| | 192 | **0.176** | **0.228** | 0.187 | 0.248 | 0.209 | 0.262 | 0.189 | 0.229 | 0.214 | 0.319 | 0.209 | 0.269 | 0.221 | 0.271 | 0.244 | 0.334 | 0.273 | 0.338 | 0.259 | 0.345 |
| | 336 | **0.185** | **0.246** | 0.193 | 0.260 | 0.209 | 0.272 | 0.197 | **0.236** | 0.235 | 0.328 | 0.222 | 0.281 | 0.217 | 0.273 | 0.259 | 0.346 | 0.266 | 0.368 | 0.273 | 0.353 |
| | 720 | **0.197** | **0.244** | 0.204 | 0.272 | 0.227 | 0.281 | 0.203 | 0.245 | 0.225 | 0.329 | 0.231 | 0.291 | 0.230 | 0.286 | 0.283 | 0.350 | 0.251 | 0.336 | 0.294 | 0.346 |
| | Avg | **0.182** | 0.233 | 0.189 | 0.255 | 0.205 | 0.265 | 0.190 | **0.232** | 0.218 | 0.317 | 0.212 | 0.272 | 0.216 | 0.268 | 0.251 | 0.337 | 0.252 | 0.338 | 0.264 | 0.336 |
| Electricity | 96 | **0.125** | **0.219** | 0.132 | 0.231 | 0.133 | 0.233 | 0.128 | 0.224 | 0.133 | 0.232 | 0.140 | 0.240 | 0.134 | 0.230 | 0.151 | 0.250 | 0.192 | 0.300 | 0.161 | 0.266 |
| | 192 | **0.142** | **0.235** | 0.150 | 0.250 | 0.157 | 0.256 | 0.147 | 0.237 | 0.148 | 0.246 | 0.158 | 0.258 | 0.154 | 0.250 | 0.165 | 0.263 | 0.202 | 0.306 | 0.169 | 0.273 |
| | 336 | **0.161** | **0.254** | 0.167 | 0.267 | 0.169 | 0.268 | 0.162 | 0.255 | 0.169 | 0.265 | 0.175 | 0.275 | 0.174 | 0.271 | 0.180 | 0.280 | 0.262 | 0.350 | 0.181 | 0.284 |
| | 720 | 0.200 | 0.288 | 0.203 | 0.299 | 0.200 | 0.292 | 0.198 | **0.286** | 0.202 | 0.295 | 0.215 | 0.310 | **0.197** | 0.292 | 0.215 | 0.313 | 0.257 | 0.349 | 0.210 | 0.308 |
| | Avg | **0.157** | **0.249** | 0.163 | 0.262 | 0.165 | 0.262 | 0.159 | 0.253 | 0.163 | 0.260 | 0.172 | 0.271 | 0.165 | 0.261 | 0.178 | 0.277 | 0.228 | 0.326 | 0.180 | 0.283 |
| Traffic | 96 | 0.373 | 0.262 | 0.362 | 0.262 | 0.380 | 0.270 | 0.358 | **0.248** | 0.385 | 0.277 | 0.393 | 0.298 | **0.346** | 0.250 | 0.428 | 0.304 | 0.556 | 0.305 | 0.489 | 0.364 |
| | 192 | 0.383 | 0.263 | 0.385 | 0.268 | 0.398 | 0.277 | 0.386 | 0.266 | 0.401 | 0.283 | 0.414 | 0.312 | **0.355** | **0.257** | 0.437 | 0.308 | 0.557 | 0.304 | 0.493 | 0.370 |
| | 336 | 0.399 | 0.270 | 0.397 | 0.275 | 0.409 | 0.289 | 0.393 | 0.263 | 0.410 | 0.287 | 0.432 | 0.322 | **0.363** | **0.265** | 0.448 | 0.314 | 0.590 | 0.312 | 0.512 | 0.383 |
| | 720 | 0.433 | **0.289** | 0.434 | 0.298 | 0.437 | 0.301 | 0.435 | 0.295 | 0.443 | 0.304 | 0.472 | 0.347 | **0.423** | **0.289** | 0.480 | 0.333 | 0.617 | 0.323 | 0.539 | 0.404 |
| | Avg | 0.397 | 0.271 | 0.395 | 0.276 | 0.406 | 0.284 | 0.393 | 0.268 | 0.410 | 0.288 | 0.428 | 0.320 | **0.372** | **0.265** | 0.448 | 0.315 | 0.580 | 0.311 | 0.508 | 0.380 |
| ETTh1 | 96 | **0.366** | **0.393** | 0.369 | 0.396 | 0.377 | 0.402 | 0.381 | 0.405 | 0.376 | 0.401 | 0.413 | 0.427 | 0.398 | 0.419 | 0.373 | 0.399 | 0.414 | 0.428 | 0.451 | 0.455 |
| | 192 | **0.394** | **0.411** | 0.410 | 0.417 | 0.404 | 0.418 | 0.416 | 0.433 | 0.410 | 0.421 | 0.449 | 0.450 | 0.436 | 0.444 | 0.407 | 0.421 | 0.469 | 0.467 | 0.482 | 0.473 |
| | 336 | 0.413 | 0.422 | **0.391** | **0.414** | 0.428 | 0.435 | 0.435 | 0.440 | 0.434 | 0.439 | 0.455 | 0.458 | 0.448 | 0.455 | 0.524 | 0.445 | 0.502 | 0.480 | 0.501 | 0.486 |
| | 720 | 0.445 | **0.453** | 0.447 | 0.464 | **0.437** | 0.459 | 0.451 | 0.460 | 0.451 | 0.471 | 0.528 | 0.521 | 0.508 | 0.508 | 0.479 | 0.501 | 0.645 | 0.542 | 0.510 | 0.509 |
| | Avg | **0.404** | **0.420** | **0.404** | 0.423 | 0.412 | 0.429 | 0.421 | 0.435 | 0.418 | 0.433 | 0.461 | 0.464 | 0.445 | 0.457 | 0.424 | 0.442 | 0.508 | 0.479 | 0.486 | 0.481 |
| ETTh2 | 96 | **0.272** | 0.333 | **0.272** | 0.334 | 0.275 | 0.336 | 0.283 | 0.346 | 0.275 | 0.337 | 0.306 | **0.326** | 0.300 | 0.357 | 0.305 | 0.368 | 0.347 | 0.395 | 0.370 | 0.431 |
| | 192 | 0.333 | 0.381 | **0.325** | **0.369** | 0.339 | 0.376 | 0.345 | 0.384 | 0.338 | 0.375 | 0.366 | 0.402 | 0.367 | 0.399 | 0.399 | 0.428 | 0.403 | 0.430 | 0.441 | 0.474 |
| | 336 | 0.346 | 0.399 | **0.319** | **0.373** | 0.360 | 0.397 | 0.372 | 0.406 | 0.352 | 0.391 | 0.400 | 0.427 | 0.392 | 0.420 | 0.471 | 0.476 | 0.441 | 0.457 | 0.469 | 0.499 |
| | 720 | **0.386** | 0.431 | 0.395 | 0.433 | 0.390 | **0.425** | 0.408 | 0.443 | 0.398 | 0.435 | 0.435 | 0.454 | 0.416 | 0.420 | 0.764 | 0.621 | 0.471 | 0.483 | 0.504 | 0.530 |
| | Avg | 0.334 | 0.386 | **0.328** | **0.377** | 0.341 | 0.384 | 0.352 | 0.395 | 0.341 | 0.385 | 0.377 | 0.412 | 0.369 | 0.405 | 0.485 | 0.473 | 0.416 | 0.441 | 0.446 | 0.484 |
| ETTm1 | 96 | **0.283** | **0.334** | 0.292 | 0.344 | 0.295 | 0.346 | 0.284 | 0.339 | 0.292 | 0.344 | 0.314 | 0.366 | 0.308 | 0.360 | 0.302 | 0.345 | 0.338 | 0.377 | 0.311 | 0.353 |
| | 192 | **0.321** | **0.359** | 0.335 | 0.371 | 0.339 | 0.371 | 0.323 | 0.363 | 0.340 | 0.373 | 0.353 | 0.389 | 0.344 | 0.382 | 0.337 | 0.368 | 0.363 | 0.386 | 0.344 | 0.372 |
| | 336 | **0.352** | **0.378** | 0.365 | 0.392 | 0.373 | 0.392 | 0.356 | 0.384 | 0.367 | 0.394 | 0.383 | 0.405 | 0.378 | 0.403 | 0.368 | 0.384 | 0.389 | 0.405 | 0.378 | 0.391 |
| | 720 | **0.399** | **0.410** | 0.417 | 0.418 | 0.420 | 0.414 | 0.411 | 0.421 | 0.422 | 0.427 | 0.437 | 0.435 | 0.439 | 0.441 | 0.422 | 0.418 | 0.514 | 0.465 | 0.433 | 0.422 |
| | Avg | **0.338** | **0.370** | 0.352 | 0.381 | 0.357 | 0.381 | 0.344 | 0.377 | 0.355 | 0.385 | 0.372 | 0.399 | 0.367 | 0.397 | 0.357 | 0.379 | 0.401 | 0.408 | 0.367 | 0.385 |
| ETTm2 | 96 | **0.161** | 0.257 | 0.170 | 0.257 | 0.165 | 0.256 | 0.172 | 0.260 | 0.166 | **0.255** | 0.179 | 0.271 | 0.181 | 0.269 | 0.165 | 0.260 | 0.217 | 0.290 | 0.189 | 0.292 |
| | 192 | **0.215** | **0.285** | 0.227 | 0.295 | 0.229 | 0.298 | 0.234 | 0.306 | 0.228 | 0.299 | 0.243 | 0.313 | 0.233 | 0.308 | 0.238 | 0.319 | 0.251 | 0.317 | 0.250 | 0.333 |
| | 336 | **0.273** | **0.325** | 0.278 | 0.327 | 0.274 | 0.328 | 0.278 | 0.335 | 0.280 | 0.331 | 0.295 | 0.345 | 0.281 | 0.338 | 0.309 | 0.370 | 0.312 | 0.358 | 0.309 | 0.372 |
| | 720 | **0.357** | **0.375** | 0.365 | 0.384 | 0.367 | 0.383 | 0.363 | 0.391 | 0.373 | 0.389 | 0.377 | 0.397 | 0.368 | 0.391 | 0.404 | 0.423 | 0.481 | 0.448 | 0.412 | 0.435 |
| | Avg | **0.252** | **0.311** | 0.260 | 0.316 | 0.259 | 0.316 | 0.262 | 0.323 | 0.262 | 0.319 | 0.273 | 0.332 | 0.266 | 0.327 | 0.279 | 0.343 | 0.315 | 0.353 | 0.290 | 0.358 |

Table 10: Multivariate long-term forecasting results compared with baseline models. **Bold** indicates the best result, and _underlined_ denotes the second best. The results are reported using input length $L = 768$ and prediction lengths $H \in \{96, 192, 336, 720\}$.

| Models | | TimePerceiver | | DeformableTST | | CARD | | CATS | | PatchTST | | iTransformer | | S-Mamba | | DLinear | | TimesNet | | TiDE | |
|---|---|---|---|---|---|---|---|---|---|---|---|---|---|---|---|---|---|---|---|---|---|
| Metric | | MSE | MAE | MSE | MAE | MSE | MAE | MSE | MAE | MSE | MAE | MSE | MAE | MSE | MAE | MSE | MAE | MSE | MAE | MSE | MAE |
| Weather | 96 | _0.144_ | 0.200 | 0.146 | _0.198_ | 0.152 | 0.208 | **0.143** | **0.197** | 0.148 | 0.202 | 0.165 | 0.217 | 0.165 | 0.219 | 0.167 | 0.226 | 0.173 | 0.228 | 0.171 | 0.225 |
| | 192 | **0.186** | **0.237** | 0.191 | _0.239_ | 0.227 | 0.269 | _0.188_ | 0.240 | 0.194 | 0.241 | 0.205 | 0.253 | 0.210 | 0.259 | 0.211 | 0.267 | 0.258 | 0.296 | 0.214 | 0.261 |
| | 336 | **0.239** | 0.283 | _0.241_ | **0.280** | 0.262 | 0.298 | **0.239** | _0.282_ | 0.243 | _0.282_ | 0.259 | 0.296 | 0.267 | 0.301 | 0.256 | 0.306 | 0.311 | 0.334 | 0.258 | 0.295 |
| | 720 | 0.316 | 0.337 | _0.310_ | _0.331_ | 0.362 | 0.356 | **0.309** | 0.333 | _0.310_ | **0.328** | 0.321 | 0.342 | 0.330 | 0.346 | 0.315 | 0.353 | 0.416 | 0.404 | 0.321 | 0.340 |
| | Avg | _0.221_ | 0.264 | 0.222 | **0.262** | 0.251 | 0.283 | **0.220** | _0.263_ | 0.224 | _0.263_ | 0.238 | 0.277 | 0.243 | 0.281 | 0.237 | 0.288 | 0.290 | 0.316 | 0.241 | 0.280 |
| Solar | 96 | **0.158** | **0.211** | 0.165 | 0.238 | 0.177 | 0.246 | _0.163_ | _0.222_ | 0.191 | 0.273 | 0.174 | 0.242 | 0.174 | 0.238 | 0.190 | 0.273 | 0.219 | 0.279 | 0.197 | 0.272 |
| | 192 | **0.180** | **0.228** | _0.184_ | 0.254 | 0.210 | 0.268 | 0.193 | _0.234_ | 0.211 | 0.282 | 0.194 | 0.259 | 0.198 | 0.264 | 0.211 | 0.291 | 0.228 | 0.297 | 0.216 | 0.293 |
| | 336 | **0.186** | **0.225** | 0.191 | 0.263 | 0.219 | 0.277 | _0.189_ | _0.247_ | 0.221 | 0.293 | 0.207 | 0.272 | 0.213 | 0.275 | 0.227 | 0.303 | 0.239 | 0.312 | 0.233 | 0.317 |
| | 720 | **0.197** | _0.253_ | 0.199 | 0.262 | 0.233 | 0.294 | _0.199_ | **0.245** | 0.235 | 0.297 | 0.217 | 0.279 | 0.217 | 0.277 | 0.234 | 0.316 | 0.238 | 0.313 | 0.237 | 0.320 |
| | Avg | **0.180** | **0.229** | _0.185_ | 0.254 | 0.210 | 0.271 | 0.186 | _0.237_ | 0.215 | 0.286 | 0.198 | 0.263 | 0.201 | 0.264 | 0.216 | 0.296 | 0.231 | 0.300 | 0.221 | 0.301 |
| Electricity | 96 | _0.128_ | _0.225_ | 0.132 | 0.234 | 0.133 | 0.230 | **0.126** | **0.222** | 0.130 | 0.235 | 0.142 | 0.241 | 0.134 | 0.232 | 0.145 | 0.243 | 0.203 | 0.310 | 0.143 | 0.250 |
| | 192 | **0.147** | **0.241** | _0.148_ | 0.248 | 0.157 | 0.256 | 0.149 | _0.243_ | 0.153 | 0.249 | 0.160 | 0.258 | 0.157 | 0.253 | 0.159 | 0.257 | 0.205 | 0.313 | 0.154 | 0.258 |
| | 336 | **0.165** | _0.260_ | **0.165** | 0.266 | 0.181 | 0.283 | 0.171 | 0.263 | _0.168_ | 0.269 | 0.179 | 0.277 | 0.169 | 0.268 | 0.174 | 0.274 | 0.228 | 0.331 | 0.176 | 0.275 |
| | 720 | 0.202 | 0.293 | 0.197 | 0.296 | 0.207 | 0.303 | _0.196_ | **0.287** | 0.205 | 0.298 | 0.221 | 0.312 | **0.194** | _0.289_ | 0.208 | 0.307 | 0.241 | 0.342 | 0.213 | 0.303 |
| | Avg | **0.160** | _0.255_ | _0.161_ | 0.261 | 0.170 | 0.268 | 0.161 | **0.254** | 0.164 | 0.263 | 0.176 | 0.272 | 0.164 | 0.264 | 0.171 | 0.270 | 0.219 | 0.324 | 0.171 | 0.272 |
| Traffic | 96 | 0.370 | 0.265 | _0.355_ | 0.261 | 0.369 | 0.267 | 0.357 | **0.248** | 0.373 | 0.267 | 0.380 | 0.291 | **0.342** | _0.250_ | 0.400 | 0.287 | 0.565 | 0.303 | 0.451 | 0.344 |
| | 192 | 0.382 | 0.268 | 0.380 | 0.271 | 0.381 | 0.270 | _0.379_ | _0.258_ | 0.384 | 0.269 | 0.400 | 0.306 | **0.362** | 0.262 | 0.411 | 0.291 | 0.579 | 0.308 | 0.459 | 0.346 |
| | 336 | 0.397 | 0.272 | _0.393_ | 0.281 | 0.402 | 0.283 | _0.393_ | _0.265_ | 0.399 | 0.275 | 0.420 | 0.317 | **0.366** | 0.268 | 0.425 | 0.298 | 0.573 | 0.304 | 0.470 | 0.353 |
| | 720 | 0.436 | _0.292_ | 0.434 | 0.300 | 0.437 | 0.299 | _0.426_ | **0.285** | 0.439 | 0.295 | 0.466 | 0.344 | **0.430** | 0.295 | 0.465 | 0.322 | 0.601 | 0.318 | 0.491 | 0.362 |
| | Avg | 0.396 | 0.274 | 0.391 | 0.278 | 0.397 | 0.280 | _0.389_ | _0.264_ | 0.399 | 0.277 | 0.417 | 0.315 | **0.375** | 0.269 | 0.425 | 0.300 | 0.579 | 0.308 | 0.468 | 0.351 |
| ETTh1 | 96 | **0.355** | **0.390** | _0.367_ | 0.404 | 0.377 | 0.410 | 0.373 | 0.407 | 0.379 | 0.410 | 0.401 | 0.428 | 0.393 | 0.425 | 0.372 | _0.401_ | 0.510 | 0.483 | 0.446 | 0.457 |
| | 192 | **0.390** | **0.415** | _0.409_ | 0.430 | 0.410 | 0.428 | 0.416 | 0.441 | 0.416 | 0.433 | 0.433 | 0.452 | 0.432 | 0.451 | 0.411 | _0.425_ | 0.537 | 0.505 | 0.483 | 0.478 |
| | 336 | **0.419** | **0.436** | _0.406_ | _0.437_ | 0.430 | 0.443 | 0.456 | 0.464 | 0.440 | 0.451 | 0.475 | 0.481 | 0.474 | 0.479 | 0.438 | 0.450 | 0.691 | 0.561 | 0.508 | 0.496 |
| | 720 | **0.455** | **0.477** | _0.458_ | _0.481_ | 0.470 | 0.487 | 0.503 | 0.489 | 0.469 | 0.483 | 0.567 | 0.549 | 0.532 | 0.528 | 0.486 | 0.508 | 0.680 | 0.556 | 0.518 | 0.519 |
| | Avg | **0.405** | **0.430** | _0.410_ | _0.438_ | 0.422 | 0.442 | 0.437 | 0.450 | 0.426 | 0.444 | 0.469 | 0.478 | 0.458 | 0.471 | 0.427 | 0.446 | 0.605 | 0.526 | 0.489 | 0.488 |
| ETTh2 | 96 | 0.278 | _0.338_ | **0.269** | **0.334** | 0.279 | 0.341 | 0.277 | 0.339 | _0.275_ | 0.339 | 0.307 | 0.363 | 0.298 | 0.356 | 0.325 | 0.386 | 0.431 | 0.446 | 0.365 | 0.427 |
| | 192 | 0.344 | 0.395 | **0.325** | **0.372** | 0.350 | 0.384 | 0.337 | 0.383 | _0.336_ | _0.378_ | 0.441 | 0.442 | 0.370 | 0.406 | 0.432 | 0.450 | 0.444 | 0.471 | 0.436 | 0.471 |
| | 336 | _0.350_ | 0.409 | **0.324** | **0.380** | 0.367 | 0.403 | 0.370 | 0.407 | 0.357 | _0.401_ | 0.452 | 0.456 | 0.394 | 0.425 | 0.535 | 0.508 | 0.513 | 0.499 | 0.459 | 0.495 |
| | 720 | _0.395_ | 0.441 | 0.412 | 0.446 | **0.392** | **0.430** | 0.435 | 0.468 | 0.397 | _0.436_ | 0.434 | 0.462 | 0.428 | 0.454 | 0.947 | 0.689 | 0.520 | 0.494 | 0.494 | 0.529 |
| | Avg | 0.342 | 0.396 | **0.333** | **0.383** | 0.347 | 0.390 | 0.355 | 0.399 | _0.341_ | _0.389_ | 0.409 | 0.431 | 0.373 | 0.410 | 0.560 | 0.508 | 0.477 | 0.483 | 0.439 | 0.480 |
| ETTm1 | 96 | **0.283** | **0.337** | 0.291 | 0.347 | 0.296 | 0.343 | _0.285_ | _0.340_ | 0.295 | 0.351 | 0.316 | 0.370 | 0.310 | 0.367 | 0.307 | 0.351 | 0.378 | 0.397 | 0.317 | 0.360 |
| | 192 | _0.324_ | **0.366** | 0.325 | 0.372 | 0.339 | 0.371 | **0.318** | _0.368_ | 0.329 | 0.373 | 0.347 | 0.387 | 0.350 | 0.390 | 0.337 | _0.368_ | 0.414 | 0.415 | 0.344 | 0.374 |
| | 336 | **0.348** | **0.379** | 0.359 | 0.390 | 0.368 | 0.388 | _0.353_ | _0.384_ | 0.364 | 0.395 | 0.388 | 0.412 | 0.377 | 0.406 | 0.364 | _0.384_ | 0.443 | 0.429 | 0.373 | 0.391 |
| | 720 | _0.410_ | **0.409** | 0.418 | 0.423 | 0.417 | 0.420 | **0.404** | 0.415 | 0.420 | 0.432 | 0.460 | 0.456 | 0.430 | 0.440 | 0.413 | _0.414_ | 0.510 | 0.474 | 0.422 | 0.418 |
| | Avg | _0.342_ | **0.373** | 0.348 | 0.383 | 0.355 | 0.381 | **0.340** | _0.377_ | 0.352 | 0.388 | 0.378 | 0.406 | 0.367 | 0.401 | 0.355 | 0.379 | 0.436 | 0.429 | 0.364 | 0.386 |
| ETTm2 | 96 | 0.166 | 0.257 | 0.169 | 0.258 | _0.164_ | _0.255_ | 0.168 | 0.262 | 0.182 | 0.272 | 0.185 | 0.278 | 0.178 | 0.271 | **0.161** | **0.253** | 0.244 | 0.314 | 0.186 | 0.291 |
| | 192 | _0.224_ | **0.293** | 0.229 | 0.299 | 0.226 | _0.298_ | 0.230 | 0.307 | 0.247 | 0.314 | 0.239 | 0.317 | 0.235 | 0.311 | **0.222** | 0.302 | 0.287 | 0.342 | 0.245 | 0.333 |
| | 336 | _0.278_ | **0.327** | 0.280 | 0.333 | **0.272** | _0.328_ | 0.283 | 0.339 | 0.303 | 0.354 | 0.298 | 0.355 | 0.290 | 0.347 | 0.296 | 0.361 | 0.326 | 0.370 | 0.304 | 0.373 |
| | 720 | _0.357_ | **0.384** | **0.349** | **0.384** | 0.390 | 0.399 | 0.367 | 0.391 | 0.371 | 0.398 | 0.401 | 0.416 | 0.362 | _0.390_ | 0.400 | 0.425 | 0.450 | 0.442 | 0.392 | 0.430 |
| | Avg | **0.256** | **0.315** | _0.257_ | _0.319_ | 0.263 | 0.320 | 0.262 | 0.325 | 0.276 | 0.335 | 0.281 | 0.341 | 0.266 | 0.330 | 0.270 | 0.335 | 0.327 | 0.367 | 0.282 | 0.357 |

# F   Full Results of Component Ablation Studies

Table 11: Full performance comparison across different formulations, encoder attention mechanisms, and positional encoding (PE) sharing strategies. **Bold** indicates the best result among all configurations. A lower MSE or MAE indicates a better performance. The first row corresponds to our proposed model. The results are reported using input length $L = 384$ and prediction lengths $H \in \{96, 192, 336, 720\}$.

| Formulation | Encoder Attention | PE Sharing | | ETTh1 MSE | ETTh1 MAE | ETTm1 MSE | ETTm1 MAE | Solar MSE | Solar MAE | ECL MSE | ECL MAE |
|---|---|---|---|---|---|---|---|---|---|---|---|
| Generalized | Latent Bottleneck | Sharing | 96 | **0.366** | **0.393** | **0.283** | **0.334** | **0.163** | **0.213** | **0.125** | **0.219** |
| | | | 192 | **0.394** | **0.411** | **0.321** | 0.359 | **0.176** | 0.228 | **0.142** | **0.235** |
| | | | 336 | **0.413** | **0.422** | 0.352 | **0.378** | **0.185** | 0.246 | 0.161 | 0.254 |
| | | | 720 | **0.445** | **0.453** | **0.399** | **0.410** | **0.197** | **0.244** | **0.200** | **0.288** |
| | | | Avg | **0.404** | **0.420** | **0.338** | **0.370** | **0.182** | 0.233 | **0.157** | **0.249** |
| Standard | Latent Bottleneck | Sharing | 96 | 0.378 | 0.403 | 0.292 | 0.346 | 0.178 | 0.228 | 0.142 | 0.239 |
| | | | 192 | 0.403 | 0.418 | 0.343 | 0.372 | 0.192 | 0.239 | 0.155 | 0.252 |
| | | | 336 | 0.426 | 0.432 | 0.373 | 0.389 | 0.198 | 0.256 | 0.173 | 0.270 |
| | | | 720 | 0.474 | 0.495 | 0.410 | 0.419 | 0.209 | 0.248 | 0.207 | 0.299 |
| | | | Avg | 0.420 | 0.437 | 0.355 | 0.382 | 0.194 | 0.243 | 0.169 | 0.265 |
| Generalized | Full Self-Attention | Sharing | 96 | 0.381 | 0.403 | 0.289 | 0.337 | 0.174 | 0.215 | 0.128 | 0.224 |
| | | | 192 | 0.406 | 0.422 | 0.341 | 0.365 | 0.188 | 0.227 | 0.150 | 0.242 |
| | | | 336 | 0.438 | 0.436 | 0.364 | 0.383 | 0.203 | 0.255 | 0.162 | 0.256 |
| | | | 720 | 0.473 | 0.475 | 0.417 | 0.416 | 0.204 | 0.251 | 0.204 | 0.292 |
| | | | Avg | 0.425 | 0.434 | 0.353 | 0.375 | 0.192 | 0.237 | 0.161 | 0.254 |
| Generalized | Decoupled Self-Attention | Sharing | 96 | 0.385 | 0.406 | 0.318 | 0.360 | 0.173 | 0.220 | 0.126 | 0.221 |
| | | | 192 | 0.411 | 0.422 | 0.331 | **0.358** | 0.184 | 0.231 | 0.146 | 0.238 |
| | | | 336 | 0.432 | 0.434 | 0.366 | 0.383 | 0.195 | **0.232** | **0.160** | **0.252** |
| | | | 720 | 0.462 | 0.468 | 0.407 | 0.413 | 0.205 | 0.247 | 0.201 | 0.290 |
| | | | Avg | 0.423 | 0.433 | 0.356 | 0.379 | 0.189 | 0.233 | 0.158 | 0.250 |
| Generalized | Full Self-Attention | Not Sharing | 96 | 0.378 | 0.398 | 0.288 | 0.341 | 0.174 | 0.215 | 0.128 | 0.220 |
| | | | 192 | 0.409 | 0.418 | 0.327 | 0.359 | 0.178 | **0.222** | 0.146 | 0.241 |
| | | | 336 | 0.439 | 0.441 | **0.350** | 0.384 | 0.205 | 0.244 | 0.169 | 0.259 |
| | | | 720 | 0.465 | 0.471 | 0.406 | 0.411 | 0.215 | 0.248 | 0.208 | 0.297 |
| | | | Avg | 0.423 | 0.432 | 0.342 | 0.374 | 0.193 | **0.232** | 0.163 | 0.254 |

As shown in Table 11, this section presents the full results of the component-wise ablation study discussed in Section 4.2. While keeping the overall architecture and generalized formulation, we further discuss the impact of encoder attention and query design in more detail.

In the encoder, to simultaneously capture both temporal and channel dependencies, we needed a method that is not only effective but also computationally efficient. We achieved this through a latent compression technique, which conceptually resembles the use of auxiliary memory in prior neural architectures that decouple computation from the input structure [29–33]. Specifically, we apply three layers of self-attention over a compact latent representation that functions as an auxiliary memory containing distilled yet essential information. This approach enables efficient and effective information extraction, and empirically demonstrates strong performance in terms of both efficiency and accuracy.

In the decoder, we previously discussed how sharing positional embeddings facilitates temporal alignment between encoder inputs and decoder queries. In our main design, we decouple the positional information by using separate channel and temporal positional embeddings (CPE and TPE). As shown in Table 12, one variation of this design combines the two into a single unified positional embedding, representing an alternative approach to query construction.

Table 12: Query design performance comparison.

| Query Design | | Decoupled PE MSE | Decoupled PE MAE | Unified PE MSE | Unified PE MAE |
|---|---|---|---|---|---|
| Metric | | | | | |
| ETTh1 | 96 | **0.366** | **0.393** | 0.379 | 0.399 |
| | 192 | **0.394** | **0.411** | 0.405 | 0.418 |
| | 336 | **0.413** | **0.422** | 0.427 | 0.433 |
| | 720 | **0.445** | **0.453** | 0.460 | 0.465 |
| | Avg | **0.404** | **0.420** | 0.418 | 0.429 |

## G  Ablation Study on Latent

**Effectiveness of Latent.** In our framework, the latent serves as an auxiliary memory [29–33] designed to capture diverse temporal-channel patterns. As shown in Table 13, the presence of the latent itself plays a key role in forecasting performance. Moreover, the performance is not highly sensitive, as long as the number of latents is reasonably large. This indicates that our latent bottleneck structure effectively aggregates temporal and channel dependencies, even with minimal capacity. We think that the latent may tend to learn redundant information. This could explain why performance does not vary significantly, especially when the latent dimension is large (*e.g.*, $D_L = 128$). To examine the hypothesis, we conduct additional configurations with different latent dimensions $D_L \in \{16, 64\}$ under the Traffic dataset. As shown in Table 14, we observe a clear performance drop as the dimension decreases.

Table 13: Performance comparison across different latent size $M \in \{0, 8, 16, 32\}$.

| Dataset | ECL | | Traffic | |
|---|---|---|---|---|
| Metric | MSE | MAE | MSE | MAE |
| 0 | 0.165 | 0.256 | 0.415 | 0.288 |
| 8 | 0.158 | **0.249** | 0.401 | 0.272 |
| 16 | 0.159 | 0.250 | 0.403 | 0.273 |
| 32 | **0.157** | **0.249** | **0.397** | **0.271** |

Table 14: Performance comparison across different latent dimension $L_D \in \{16, 64, 128\}$ with latent size $M = 1$ on traffic dataset. **Bold** indicates the best result. A lower MSE or MAE indicates a better performance. The results are reported using input length $L = 384$ and prediction lengths $H \in \{96, 192, 336, 720\}$

| Latent Dimension | 16 | | 64 | | 128 | |
|---|---|---|---|---|---|---|
| Metric | MSE | MAE | MSE | MAE | MSE | MAE |
| 96 | 0.392 | 0.271 | 0.385 | 0.266 | **0.378** | **0.264** |
| 192 | 0.392 | 0.265 | 0.394 | **0.262** | **0.388** | 0.263 |
| 336 | 0.415 | 0.284 | 0.411 | 0.282 | **0.403** | **0.276** |
| 720 | 0.436 | 0.294 | 0.435 | **0.290** | **0.433** | **0.290** |
| Avg | 0.408 | 0.279 | 0.406 | 0.275 | **0.401** | **0.273** |

## H  Imputation

Table 15: Performance comparison on the imputation task with sequence length 192. The masking is applied in units of patch length 24, and experiments are conducted by masking either one or two patches.

| Dataset | Masked Patch | TimePerceiver | | TimesNet | | iTransformer | | PatchTST | |
|---|---|---|---|---|---|---|---|---|---|
| | | MSE | MAE | MSE | MAE | MSE | MAE | MSE | MAE |
| ETTh1 | 1 | 0.218 | 0.305 | 0.301 | 0.356 | 0.257 | 0.328 | 0.241 | 0.313 |
| | 2 | 0.248 | 0.326 | 0.309 | 0.362 | 0.267 | 0.341 | 0.266 | 0.331 |
| Weather | 1 | 0.066 | 0.093 | 0.072 | 0.104 | 0.066 | 0.094 | 0.061 | 0.081 |
| | 2 | 0.078 | 0.109 | 0.090 | 0.126 | 0.073 | 0.109 | 0.069 | 0.102 |
| ECL | 1 | 0.097 | 0.200 | 0.120 | 0.233 | 0.093 | 0.196 | 0.145 | 0.259 |
| | 2 | 0.106 | 0.205 | 0.123 | 0.234 | 0.102 | 0.206 | 0.156 | 0.271 |

We conduct imputation experiments with three datasets (ETTh1, Weather, Electricity) and three baselines [11, 6, 4]. We here set a more challenging task of imputation via patch-wise masking with a patch length of 24, as opposed to the easier setup of masking at individual timesteps. As shown in Table 15, TIMEPERCEIVER consistently achieves strong performance across all datasets, even though **we did not specifically tailor our framework to the imputation task**.

Table 16: Performance comparison with more baselines which are MLP-based, CNN-based and SSM-based models. **Bold** indicates the best result. A lower MSE or MAE indicates a better performance. Results are the average performance over four prediction lengths $H \in \{96, 192, 336, 720\}$ with input length $L = 96$.

| Models | TimePerceiver | | TimeMixer | | CycleNet | | ModernTCN | | Sor-Mamba | |
|---|---|---|---|---|---|---|---|---|---|---|
| Metric | MSE | MAE | MSE | MAE | MSE | MAE | MSE | MAE | MSE | MAE |
| Weather | **0.238** | **0.269** | 0.240 | 0.271 | 0.243 | 0.271 | 0.240 | 0.274 | 0.256 | 0.277 |
| ECL | **0.165** | **0.258** | 0.178 | 0.272 | 0.168 | 0.259 | 0.204 | 0.286 | 0.168 | 0.264 |
| Traffic | 0.428 | 0.275 | 0.484 | 0.297 | 0.472 | 0.313 | 0.625 | 0.376 | **0.402** | **0.273** |
| ETTh1 | **0.422** | **0.427** | 0.447 | 0.440 | 0.432 | **0.427** | 0.436 | 0.429 | 0.433 | 0.436 |
| ETTh2 | **0.355** | **0.395** | 0.364 | 0.395 | 0.383 | 0.404 | 0.358 | 0.398 | 0.376 | 0.405 |
| ETTm1 | **0.362** | **0.383** | 0.381 | 0.395 | 0.379 | 0.395 | 0.389 | 0.403 | 0.391 | 0.400 |
| ETTm2 | 0.276 | 0.316 | 0.275 | 0.323 | **0.266** | **0.314** | 0.280 | 0.323 | 0.281 | 0.327 |

Table 17: Performance comparison with more baselines which are Graph-based and RNN-based models. **Bold** indicates the best result. A lower MSE or MAE indicates a better performance. Results are the average performance over four prediction lengths $H \in \{96, 192, 336, 720\}$ with input length $L = 96$. '-' denotes result is not reported in the official publication.

| Models | TimePerceiver | | SageFormer | | CrossGNN | | MSGNet | | SegRNN | | xLSTM-Mixer | |
|---|---|---|---|---|---|---|---|---|---|---|---|---|
| Metric | MSE | MAE | MSE | MAE | MSE | MAE | MSE | MAE | MSE | MAE | MSE | MAE |
| Weather | **0.238** | **0.269** | 0.247 | 0.273 | 0.247 | 0.289 | 0.249 | 0.278 | 0.252 | 0.299 | 0.254 | 0.275 |
| ECL | **0.165** | **0.258** | 0.175 | 0.273 | 0.201 | 0.271 | 0.194 | 0.300 | 0.184 | 0.278 | 0.174 | 0.259 |
| Traffic | **0.428** | 0.275 | 0.436 | 0.285 | 0.583 | 0.323 | - | - | 0.651 | 0.324 | 0.430 | **0.256** |
| ETTh1 | **0.422** | **0.427** | 0.431 | 0.433 | 0.437 | 0.434 | 0.452 | 0.452 | 0.425 | 0.429 | 0.442 | 0.430 |
| ETTh2 | **0.355** | **0.395** | 0.374 | 0.403 | 0.363 | 0.418 | 0.396 | 0.417 | 0.374 | 0.406 | 0.377 | 0.402 |
| ETTm1 | **0.362** | **0.383** | 0.388 | 0.400 | 0.393 | 0.404 | 0.398 | 0.411 | 0.388 | 0.405 | 0.386 | 0.389 |
| ETTm2 | **0.276** | 0.316 | 0.277 | 0.322 | 0.282 | 0.330 | 0.288 | 0.330 | 0.278 | 0.324 | 0.277 | **0.314** |

# I  Comparison with Other Baselines

In Table 1, we cover diverse model families (Transformer [4, 6–9], SSM [17], CNN [11], and MLP [13, 14]), while placing greater emphasis on Transformer-based baselines because our method is Transformer-based. As shown in Table 16, to complement this focus, we add comparisons with recent non-Transformer models, including SSM [18], CNN [34], and MLP [35, 36] based models. In additional comparisons with recent SOTA methods, our model achieves the best performance in 10 out of 14 cases.

As shown in Table 17, we here include more comparisons with recently proposed graph-based and RNN-based models that have attracted attention due to their novel foundations. Specifically, we compare against graph-based models such as SageFormer [37], CrossGNN [38], and MSGNet [39] as well as RNN-based models including SegRNN [40] and xLSTM-Mixer [41]. Our model still demonstrates superior performance in 12 out of 14 cases.

## J  Discussion

**Limitations.** While this work presents a flexible and unified framework for time-series forecasting and demonstrates superior performance over recent baselines, it primarily focuses on multivariate forecasting benchmarks with regular time intervals, following common practice in the literature. Given that our framework is capable of modeling arbitrary time segments via latent bottlenecks, we believe it can be extended to irregular or event-based time series, which is an important direction for future work.

**Broader Impacts.** This work presents a new formulation for time series modeling that broadens its applicability beyond traditional forecasting tasks. Unlike conventional methods that predict the future based only on past observations, our approach allows for flexible prediction across different temporal segments, including inferring from future to past or estimating intermediate time points. This makes it applicable to a variety of real-world scenarios, such as reconstructing unobserved data. Therefore, we strongly believe our framework can be applied to diverse domains, such as healthcare, finance, and environmental monitoring, where time series data plays a critical role, and can inspire future research toward unified and generalizable forecasting systems that better reflect the diversity of real-world applications.

**Ethics Considerations.** As our work focuses on core methodological contributions rather than specific applications, we do not foresee immediate ethical concerns.

