# OpenReview forum: "TimePerceiver: An Encoder-Decoder Framework for Generalized Time-Series Forecasting"
_NeurIPS.cc/2025/Conference — NeurIPS 2025 poster_

### Official Review · Reviewer_QYcP · 2025-06-24

**Clarity:** 3
**Significance:** 2
**Originality:** 3
**Rating:** 4
**Confidence:** 5

**Summary:**

The authors propose a new attention based encoder-decoder architecture along with a new training regime to unify the development of encoder and decoder as a single task (Generalized Formulation). This helps the encoder to attend to multiple arbitrary sampled segments of the input. Their experiments show that they outperform various (transformer focused) baselines on the standard multivariate long-term forecasting benchmark datasets. Furthermore, the authors propose a novel latent bottleneck  that compresses input patches into a small set of learnable tokens to improve from O(n^2) to O(NM), with M << N.

**Questions:**

- Q1. How does your architecture differ from T5? Please discuss how your generalized formulation can be projected to other architectures (or is this impossible?).

- Q2. In Table 2, could the authors provide all combinations? It is unclear what impact positional embedding sharing has on the standard formulation.

- Q3. While the focus is on transformer-based methods, a global view should include other architectures. Please provide results for modern RNN methods (xLSTM-Mixer), MLPs (CycleNet, TimeMixer, TimeMixer++), and convolutional models (ModernTCN).

- Q4. In your ablation, you report that ETTh1 works best for contiguous sampling. Could the authors provide a rationale? Simply stating that this highlights the importance of sampling ratio is insufficient, especially since one of the main claims is that previous pretraining methods do not model this behavior correctly.

**Ethical Concerns:**

["NO or VERY MINOR ethics concerns only"]

**Final Justification:**

I've raised my score to a borderline accept as the authors answered all my remaining concerns, added missing baselines, and showcased how the components interplay with each other. Furthermore, they provided clarifications and added missing ablations. However, due to the still missing "final" baseline results and the sheer amount of changes that will be added to the paper, I am not comfortable going higher than a borderline accept. Otherwise, this is strong work and could be published at some point at NeurIPS or a similar venue.

**Limitations:**

yes

**Paper Formatting Concerns:**

- The use of citation ranges (e.g., [13–15], [4–18]) is problematic: readers, search engines, and even LLMs cannot quickly interpret multiple works cited in this way.

- Marking individual research questions and referring back to them later would improve readability.

**Quality:**

1

**Strengths And Weaknesses:**

- Strengths:

    - The paper introduces a novel unified encoder-decoder framework and an interesting idea of sampling multiple temporal segments.

    - The figures are clear and aid understanding of the architecture
    - The manuscript is easy to follow.
    - The experiments are extensive, although focused on long-term forecasting.

    - The authors outline research questions, which improves the structure.

- Weaknesses:

    - The authors omit recent architectures (e.g., CNN- and MLP-based models such as CycleNet, TimeMixer, TimeMixer++, and ModernTCN), as well as entire model families (RNNs) and newly introduced (x)LSTMs (xLSTM-Mixer, Tirex, SegRNN), all of which achieve SOTA results in time series forecasting.

    - Masking and reconstruction tasks are common pretraining strategies but not the only ones; prior time series pretraining works include TS-TCC, TFC, TS2Vec, and others that utilize multiple self-supervised losses.

    - The significance and novelty of the architecture are unclear, as the work appears to build directly on previous pretraining methods while ignoring modern foundation models that already use an encoder-decoder architecture (for example, Lag-Llama and Chronos on top of the T5 architecture). In general, the related work section is not exhaustive. The authors should include foundation models such as TimesFM (decoder-only) and Chronos or Chronos-Bolt (patch-based), as well as some of the earlier time series pretraining works.

    - While the unified view is appealing, it is confusing how this relates to general time series model analysis; if a unified view is employed, methods from imputation should also be included in the evaluation or the related work section.

    - The experiments over different input lengths are appreciated, but results without multiple seeds are less meaningful.

    - The notation for the generalized framework is confusing; equations 5–7 require multiple readings to understand.

    - The number of latent tokens is not sufficiently explored; reporting that 8, 16, and 32 tokens work best is not enough. What happens when decreasing or increasing this number? From a practical perspective, it is unclear how to choose this hyperparameter.


### Refs

Kraus, Maurice, et al. "xLSTM-Mixer: Multivariate Time Series Forecasting by Mixing via Scalar Memories." _arXiv preprint arXiv:2410.16928_ (2024).

Auer, Andreas, et al. "TiRex: Zero-Shot Forecasting Across Long and Short Horizons with Enhanced In-Context Learning." _arXiv preprint arXiv:2505.23719_ (2025).

Lin, Shengsheng, et al. "Segrnn: Segment recurrent neural network for long-term time series forecasting." _arXiv preprint arXiv:2308.11200_ (2023).

Eldele, Emadeldeen, et al. "Time-series representation learning via temporal and contextual contrasting." IJCAI (2021)

Zhang et al. "Self-Supervised Contrastive Pre-Training For Time Series via Time-Frequency Consistency" NeurIPS (2022)

Yue, Zhihan, et al. "Ts2vec: Towards universal representation of time series." _Proceedings of the AAAI conference on artificial intelligence_. Vol. 36. No. 8. 2022.

Ansari, Abdul Fatir, et al. "Chronos: Learning the language of time series." _arXiv preprint arXiv:2403.07815_ (2024).

Das, Abhimanyu et al. "A decoder-only foundation model for time-series forecasting" ICML (2024)

Rasul, Kashif, et al. "Lag-llama: Towards foundation models for time series forecasting." _R0-FoMo: Robustness of Few-shot and Zero-shot Learning in Large Foundation Models_. 2023.

Lin, Shengsheng, et al. "Cyclenet: Enhancing time series forecasting through modeling periodic patterns." _Advances in Neural Information Processing Systems_ 37 (2024): 106315-106345.

Wang, Shiyu, et al. "Timemixer: Decomposable multiscale mixing for time series forecasting." ICLR (2024).

Wang, Shiyu, et al. "Timemixer++: A general time series pattern machine for universal predictive analysis." ICLR (2025).

Luo, Donghao, and Xue Wang. "Moderntcn: A modern pure convolution structure for general time series analysis." ICLR (2024).

---

> ### Author Rebuttal · Authors · 2025-07-30
>
> Dear Reviewer QYcP,
>
> We sincerely thank you for your helpful feedback and insightful comments. In what follows, we address your concerns one by one. Note that we only report MSE metrics in this rebuttal due to its limited space.
>
> ---
>
> **[Q1-1, W3]** How does your architecture differ from T5?
>
> **[Response]**  TimePerceiver is fundamentally different from Chronos-T5 in the perspectives about how the model operates based on the context of time. Firstly, TimePerceiver consists of a decoder with query tokens containing time and channel information about target segments. Therefore, the decoder flexibly extracts information based on the time point we want to predict from the representation captured by the encoder through the latent bottleneck, which captures temporal-channel dependencies.
> However, as stated in the Chronos paper, Chronos-T5 completely omits time and frequency information and treats time series as a plain sequence of tokens. The decoder works in an auto-regressive manner, looking only at the tokens generated so far and producing the probability distribution for the next single token. As a result, the model relies purely on sequential token patterns to predict the next value, without any position-specific information that is unique to time-series data.
>
> To sum up, TimePerceiver can respond flexibly even if the input and output intervals vary each time. On the other hand, Chronos-T5 remains stuck in a token-next-token approach, making it difficult to structurally solve generalized formulation other than predicting specific future intervals.
>
> ---
>
> **[Q1-2]** Please discuss how your generalized formulation can be projected to other architectures (or is this impossible?).
>
> **[Response]** We believe that our generalized formulation can be easily projected to other patch-based architectures by replacing their prediction layers with our query-based decoder. PatchTST [1] is a clear example: replacing its linear decoder by our query-based decoder and adopting it with our generalized formulation yields a TimePerceiver variant that uses PatchTST’s encoder design. Not only that, but the more naturally a structure can handle temporal positional information, the more easily our idea can be projected.
>
> [1] Nie et al., A Time Series is Worth 64 Words: Long-term Forecasting with Transformers. ICLR 2023
>
> ---
>
> **[Q2-1]** In Table 2, could the authors provide all combinations?
>
> **[Response]** Below are the results for the combinations that were not shown for the standard formulation in Experiment 4.2 and Appendix D.
>
> | Formulation | Encoder Attention | ETTh1 | ETTm1 | Solar | ECL |
> | :---- | :---- | :---- | :---- | :---- | :---- |
> | Standard | Full Self-Attention | 0.437 | 0.368 | 0.194 | 0.173 |
> | Standard | Decoupled Self-Attention | 0.445 | 0.377 | 0.196 | 0.166 |
>
> ---
>
> **[Q2-2]** It is unclear what impact positional embedding sharing has on the standard formulation.
>
> **[Response]** In the standard formulation, PE sharing means that channel positional embedding (CPE) is shared across the encoder and the decoder in our architecture. In contrast, our generalized formulation, PE sharing means that both CPE and temporal positional embedding (TPE) are shared. Therefore, PE sharing has less impact in the standard formulation, as we observed a slight performance difference in its ablation study. We will clarify the ablation experiments in the final manuscript.
>
> ---
>
> **[Q3, W1]** While the focus is on transformer-based methods, a global view should include other architectures. Please provide results for modern RNN methods (xLSTM-Mixer), MLPs (CycleNet, TimeMixer, TimeMixer++), and convolutional models (ModernTCN).
>
> **[Response]** We provide comparison results for the methods you mentioned below. All evaluations were conducted with a lookback window L=96 for consistency. Since public results and scripts for xLSTM-Mixer under this setting are not available, we instead include SegRNN for comparison.
>
> Experimental results show that TimePerceiver consistently outperforms the baselines. While TimeMixer++ is reported to perform slightly better than our method, it is worth noting that the reported standard deviation in the original paper is relatively high (around 0.2 across most datasets), and the code has not been publicly released. Moreover, when we run experiments using an unofficially released version of the code, we were unable to reproduce the reported results. For these reasons, we are cautious about making a direct comparison with TimeMixer++.
>
> | Dataset | TimePerceiver | TimeMixer++ | TimeMixer | CycleNet | SegRNN | ModernTCN |
> | :--- | :--- | :--- | :--- | :--- | :--- | :--- |
> | Weather | 0.238±0.002 | 0.226±0.008 | 0.240 | 0.243 | 0.252 | 0.240 |
> | ECL | 0.165±0.003 | 0.165±0.017 | 0.178 | 0.168 | 0.184 | 0.204 |
> | Traffic | 0.428±0.008 | 0.416±0.027 | 0.484 | 0.472 | 0.651 | 0.625 |
> | ETTh1 | 0.422±0.005 | 0.419±0.023 | 0.447 | 0.432 | 0.425 | 0.436 |
> | ETTh2 | 0.355±0.007 | 0.339±0.020 | 0.364 | 0.383 | 0.374 | 0.358 |
> | ETTm1 | 0.362±0.003 | 0.369±0.019 | 0.381 | 0.379 | 0.388 | 0.389 |
> | ETTm2 | 0.276±0.002 | 0.269±0.021 | 0.275 | 0.266 | 0.278 | 0.280 |
>
> ---
>
> **[Q4]** In your ablation, you report that ETTh1 works best for contiguous sampling. Could the authors provide a rationale? Simply stating that this highlights the importance of sampling ratio is insufficient, especially since one of the main claims is that previous pretraining methods do not model this behavior correctly.
>
> **[Response]** Based on our visualization and analysis of the datasets, we observed that (i) fully contiguous sampling is effective when capturing global patterns is important for the data within a single instance, whereas (ii) fully disjoint sampling performs better when the data contains frequently repeated local patterns.
> First, fully contiguous sampling proved effective not only for ETTh1 but also for the Weather dataset. Both datasets exhibit strong global patterns rather than repetitive local behaviors. In such cases, removing long contiguous segments from the input forces the model to infer high-level temporal dependencies, which helps in learning global trends more effectively.
> Second, datasets such as Solar, ECL, and Traffic showed more consistent global patterns but contained local variations, often with repetitive short-term patterns. For these datasets, fully disjoint sampling led to better performance, as it encourages the model to solve many small-scale reconstruction tasks and thus better capture local dynamics. Therefore, when prior knowledge about the characteristics of the data is available, we recommend selecting a sampling strategy accordingly to best exploit that knowledge. In the absence of such prior information, using the mixed version serves as a robust default choice. We will incorporate this clarification into the revision.
>
> ---
>
> **[W2]** Masking and reconstruction tasks are common pretraining strategies but not the only ones; prior time series pretraining works include TS-TCC, TFC, TS2Vec, and others that utilize multiple self-supervised losses.
>
> **[Response]** Our contribution lies not only in the generalized forecasting formulation but also in the encoder-decoder architecture that is closely aligned with this formulation. Furthermore, we believe that our framework is compatible with various self-supervised learning objectives mentioned (e.g., contrastive learning) and can be extended to incorporate them effectively.
>
> ---
>
> **[W4]** While the unified view is appealing, it is confusing how this relates to general time series model analysis; if a unified view is employed, methods from imputation should also be included in the evaluation or the related work section.
>
> **[Response]** Following your suggestion, we conduct imputation experiments with three datasets (ETTh1, Weather, Electricity) and three baselines (TimesNet, iTransformer, PatchTST). We here set a more challenging task of imputation via patch-wise masking with a patch length of 24, as opposed to the easier setup of masking at individual timesteps. Due to the space limit, please refer to [Reviewer DDKH, W3 Response] for the imputation results. As shown in the results, our TimePerceiver consistently outperforms the baselines across all datasets. We will add these results with more baselines into the final manuscript.
>
> ---
>
> **[W5]** The experiments over different input lengths are appreciated, but results without multiple seeds are less meaningful.
>
> **[Response]** In Appendix B, we clarify that all reported results are the average of 5 runs using different random seeds. Also, across all 32 experimental settings with lookback window L=384, 25 cases exhibit a standard deviation less than or equal to 0.0005, indicating that TimePerceiver remains highly stable. For details, refer to [Reviewer DDKH, W1 Response].
>
> ---
>
> **[W6]** The number of latent tokens is not sufficiently explored; reporting that 8, 16, and 32 tokens work best is not enough. What happens when decreasing or increasing this number? From a practical perspective, it is unclear how to choose this hyperparameter.
>
> **[Response]** Thank you for mentioning the important point. In our experiments, we simply chose the latent size based on validation. Following your suggestion, we further explore its sensitivity as reported below. We observe that the presence of a latent bottleneck clearly improves performance over having none, but the specific number of latent tokens plays only a minor role in determining the final performance, as they are all relatively small compared to the number of input tokens. This indicates that our model is not overly sensitive to this parameter, and even with a small latent size, we observe substantial performance gains.
>
> | Latent Size | ECL | Traffic |
> | :--- | :--- | :--- |
> | 0 | 0.165 | 0.415 |
> | 8 | 0.158 | 0.401 |
> | 16 | 0.159 | 0.404 |
> | 32 | 0.158 | 0.405 |
>
> ---

---

> > ### Comment · Reviewer_QYcP · 2025-08-04
> >
> > I really thank the authors for their exhaustive responses.
> >
> > [Q1-1, W3] I understand now! This clarification helped me grasp the differences. Thank you.
> >
> > [Q1-2] That’s good to hear. As this makes your work more significant, do you by any chance have a few preliminary results for this projection?
> >
> > [Q2-1] Thank you; this makes your components much clearer.
> >
> > [Q2-2] Thank you.
> >
> > [Q3, W1]:
> > -  I would've appreciated it if you provided all results. Similar to your response to **pmxb** and **OFA**, the response is not satisfactory. Not all results/configurations will ever be available for every method, but a small hyper-parameter search should be the least we, as authors, can do to combine the results of different methods. Especially when they claim to be SOTA (I mean xLSTM-Mixer in this case, not your method).
> >
> > - Furthermore, in favor of your method, could you provide your recomputed TimeMixer++ results?
> >
> > - On which datasets did your method perform better than the other baselines? Did it significantly outperform them in a few scenarios (e.g., horizons 96 and 336), or was it only marginally better across most horizons? I acknowledge that these methods sometimes improve forecasting results only marginally, but I want to understand where the strengths of your method lie (e.g. long vs short horizon).
> >
> > [Q4] Very interesting to hear—thank you.
> >
> > [W4] I asked my question directly on the reviewer thread.
> >
> > [W5] Thank you! It seems I missed this information!
> >
> > [W6] I would assume a latent size of 1 yields the same outcome as 0. When does the latent-size improvement start to take effect? Is 8 the smallest value that shows a benefit?

---

> ### Author Response · Authors · 2025-08-05
> **Additional response to Reviewer QYcP [1/3]**
>
> Dear Reviewer QYcP,
>
> Thank you for your comments and for taking the time to review our manuscript. We are happy that our rebuttal and experiments addressed most of your concerns.
>
> We here provide an additional response to [Q1-2], [Q3/W1] and [W6] for addressing your remaining concerns.
>
> ---
>
> **[Q1-2] Do you by any chance have a few preliminary results for this projection?** \
> Yes. In the early stage of our research project, we conducted a few preliminary results showing that our generalized formulation can improve other encoders. We here provide the results obtained using PatchTST as an example, under ETTh1 with a lookback window L=384. As shown below, we observed that training with the generalized formulation consistently outperforms the standard formulation across various lookback window settings. While the TimePerceiver achieves the highest performance overall, these results suggest that the benefits of the generalized formulation are not limited to a specific model and can be effectively applied to enhance various encoder architectures.
>
> | Input Length | TimePerceiver (generalized) | PatchTST (generalized) | PatchTST (standard) |
> | :---- | :---- | :---- | :---- |
> | 96 | 0.366 | 0.373 | 0.383 |
> | 192 | 0.394 | 0.403 | 0.408 |
> | 336 | 0.413 | 0.426 | 0.433 |
> | 720 | 0.445 | 0.458 | 0.467 |
> | Avg | 0.404 | 0.415 | 0.423 |

---

> ### Author Response · Authors · 2025-08-05
> **Additional response to Reviewer QYcP [2/3]**
>
> **[Q3/W1] More comparisons with recent baselines** \
> Following your request, we conduct additional experiments with xLSTM-Mixer and TimeMixer++ on ETTh1, ETTh2, ETTm1, and ETTm2 with various lookback window sizes  (L=96, 384) and forecasting horizons (H=96,192,336,720). As shown below, our TimePerceiver consistently outperforms both xLSTM-Mixer and TimeMixer++ across these diverse settings.
>
> | ETTh1 | TimePerceiver (L=96) | TimeMixer++ (L=96) | xLSTM-Mixer (L=96) | TimePerceiver (L=384) | TimeMixer++ (L=384) | xLSTM-Mixer (L=384) |
> | :--- | :--- | :--- | :--- | :--- | :--- | :--- |
> | H=96 | 0.372 | 0.384 | 0.384 | 0.366 | 0.396 | 0.369 |
> | H=192 | 0.422 | 0.433 | 0.442 | 0.394 | 0.447 | 0.413 |
> | H=336 | 0.444 | 0.481 | 0.474 | 0.413 | 0.556 | 0.423 |
> | H=720 | 0.451 | 0.474 | 0.469 | 0.445 | 0.584 | 0.438 |
> | Avg | 0.422 | 0.443 | 0.442 | 0.404 | 0.496 | 0.411 |
>
> | ETTh2 | TimePerceiver (L=96) | TimeMixer++ (L=96) | xLSTM-Mixer (L=96) | TimePerceiver (L=384) | TimeMixer++ (L=384) | xLSTM-Mixer (L=384) |
> | :--- | :--- | :--- | :--- | :--- | :--- | :--- |
> | H=96 | 0.285 | 0.299 | 0.290 | 0.272 | 0.293 | 0.280 |
> | H=192 | 0.365 | 0.369 | 0.371 | 0.333 | 0.393 | 0.348 |
> | H=336 | 0.366 | 0.431 | 0.420 | 0.346 | 0.382 | 0.367 |
> | H=720 | 0.404 | 0.451 | 0.427 | 0.386 | 0.407 | 0.395 |
> | Avg | 0.355 | 0.388 | 0.377 | 0.334 | 0.369 | 0.348 |
>
> | ETTm1 | TimePerceiver (L=96) | TimeMixer++ (L=96) | xLSTM-Mixer (L=96) | TimePerceiver (L=384) | TimeMixer++ (L=384) | xLSTM-Mixer (L=384) |
> | :--- | :--- | :--- | :--- | :--- | :--- | :--- |
> | H=96 | 0.306 | 0.320 | 0.322 | 0.283 | 0.309 | 0.278 |
> | H=192 | 0.343 | 0.361 | 0.361 | 0.321 | 0.343 | 0.323 |
> | H=336 | 0.371 | 0.382 | 0.393 | 0.352 | 0.399 | 0.363 |
> | H=720 | 0.428 | 0.450 | 0.467 | 0.399 | 0.451 | 0.417 |
> | Avg | 0.362 | 0.378 | 0.386 | 0.338 | 0.376 | 0.345 |
>
> | ETTm2 | TimePerceiver (L=96) | TimeMixer++ (L=96) | xLSTM-Mixer (L=96) | TimePerceiver (L=384) | TimeMixer++ (L=384) | xLSTM-Mixer (L=384) |
> | :--- | :--- | :--- | :--- | :--- | :--- | :--- |
> | H=96 | 0.170 | 0.179 | 0.171 | 0.161 | 0.180 | 0.162 |
> | H=192 | 0.236 | 0.255 | 0.238 | 0.215 | 0.264 | 0.219 |
> | H=336 | 0.298 | 0.315 | 0.303 | 0.273 | 0.287 | 0.273 |
> | H=720 | 0.399 | 0.421 | 0.395 | 0.357 | 0.373 | 0.366 |
> | Avg | 0.276 | 0.293 | 0.277 | 0.252 | 0.276 | 0.255
>
> Based on these results, **our TimePerceiver outperforms all the baselines you mentioned** (CycleNet, TimeMixer, TimeMixer++, xLSTM-Mixer, ModernTCN). This demonstrates the superiority of TimePerceiver over recent state-of-the-art methods. We will include these results in the final manuscript.
>
> **Note:** We are currently reproducing the results of TimeMixer++ and xLSTM-Mixer on the remaining datasets (Weather, Solar, ECL, Traffic), and we will share them as soon as they become available.
>
> ---
>
> **[Q3/W1] On which datasets or settings did your method outperform other baselines?** \
> As shown in Appendix C, TimePerceiver demonstrates consistently strong performance across a wide range of settings. It achieves competitive or superior results in various horizons, which highlights the robustness of our framework across diverse scenarios.
>
> Also, when combined with the discussion in **[Q3, W1]**, our method shows remarkable stability in datasets with relatively long sequences such as ETTm1, ETTm2, Weather and ECL. The standard deviation in these settings is notably low, indicating stable results. We attribute this to our generalized formulation, which enables the model to make more robust predictions in diverse scenarios, especially when the datasets contain large numbers of instances.

---

> ### Author Response · Authors · 2025-08-05
> **Additional response to Reviewer QYcP [3/3]**
>
> **[W6] I would assume a latent size of 1 yields the same outcome as 0. When does the latent-size improvement start to take effect? Is 8 the smallest value that shows a benefit?** \
> As shown below, interestingly, even a single latent is sufficient to yield significant improvements over the no-latent case, suggesting that the presence of the latent itself plays a key role in forecasting performance. Moreover, the performance is not highly sensitive to the number of latents. This indicates that our latent bottleneck structure effectively aggregates temporal and channel dependencies, even with minimal capacity. We appreciate the suggestion to conduct this ablation study, and we believe that the resulting observations further strengthen the contribution of our latent bottleneck. We will include these results in the final manuscript.
>
> | Latent Size | ECL | Traffic |
> | :---: | :---: | :---: |
> | 0 | 0.165 | 0.415 |
> | 1 | 0.159 | 0.403 |
> | 2 | 0.158 | 0.403 |
> | 4 | 0.158 | 0.402 |
> | 8 | 0.158 | 0.401 |
> | 16 | 0.159 | 0.404 |
> | 32 | 0.158 | 0.405 |
>
> ---
>
> Please do not hesitate to let us know if there are any remaining concerns or clarifications you would like us to address. We are more than willing to make the most of the remaining discussion time to further improve our paper.
>
> Thank you very much, \
> Authors

---

> > ### Comment · Reviewer_QYcP · 2025-08-05
> >
> > You answered all of my remaining concerns! I’m really thankful for your time.
> > That will be my last follow up question!
> >
> > That’s interesting to see. However, I have a hard time understanding how this parameter yield identical results no matter its configuration? Is the information present in a redundant way? Why doesn’t it start to regularize the signal?

---

> ### Author Response · Authors · 2025-08-05
>
> Dear Reviewer QYcP,
>
> Thank you for your follow-up question! We appreciate the opportunity to clarify further.
>
> We think that the latent may tend to learn redundant information as you mentioned. This could explain why performance does not vary significantly, especially when the latent dimension is large (e.g., dim=128 in the previous experiments). To examine the hypothesis, we conduct additional configurations with different latent dimensions (dim=16, 64) under the Traffic dataset. As shown below, we observe a clear performance drop as the dimension decreases.
>
> |  | 1x16 | 1x64 | 1x128 |
> | :--- | :--- | :--- | :--- |
> | H=96 | 0.392 | 0.385 | 0.388 |
> | H=192 | 0.392 | 0.394 | 0.388 |
> | H=336 | 0.412 | 0.411 | 0.403 |
> | H=720 | 0.436 | 0.435 | 0.433 |
> | Avg | 0.408 | 0.406 | 0.403 |
>
> Furthermore, we compare two different latent configurations with the same latent capacity (i.e., number of latent x latent dim): 4x16 vs 1x64. As shown below, interestingly, we observe that using more latent tokens tends to perform better. This suggests that **the latent structure plays an important role in our framework.**
>
> |  | 4x16 | 1x64 |
> | :--- | :--- | :--- |
> | H=96 | 0.381 | 0.385 |
> | H=192 | 0.392 | 0.394 |
> | H=336 | 0.402 | 0.411 |
> | H=720 | 0.433 | 0.435 |
> | Avg | 0.402 | 0.406 |
>
> Although finding the optimal latent configuration remains an open problem, we would like to emphasize that **our latent bottleneck achieves strong performance over the no-latent case even without carefully tuned configurations.** We again appreciate your comment, which helps further strengthen our paper’s contribution. We will include this new ablation study in the final manuscript.
>
> Thank you very much, \
> Authors

---

> ### Comment · Reviewer_QYcP · 2025-08-05
>
> I thank the authors again for their engagement and elaboration. For the camera-ready/revision, the authors can think about plotting those (dim-reduced) latent tokens to give a qualitative intuition of what is happening for different configurations.
> Otherwise, I have no remaining concerns.
>
> I raised my score accordingly.

---

> ### Author Response · Authors · 2025-08-05
>
> Dear Reviewer QYcP,
>
> Thank you very much for your thoughtful comments and for taking the time to review our manuscript. We are glad that our additional clarifications and experiments have addressed your concerns. We will incorporate all the findings and the discussions from the rebuttal into the final manuscript.
>
> If you have any further suggestions, we would be happy to hear them.
>
> Sincerely, \
> Authors

---

### Official Review · Reviewer_pmxb · 2025-06-26

**Clarity:** 3
**Significance:** 3
**Originality:** 2
**Rating:** 3
**Confidence:** 5

**Summary:**

The author propose TimePerceiver, a generalized Transformer structure for time series forecasting. The authors propose three key innovations: latent bottleneck representations-based encoder, learnable query-based decoder and a hybrid architecture supporting flexible sequence length forecasting.

Experiments on benchmarks (e.g., ETT, Traffic, Weather) demonstrate state-of-the-art (SOTA) performance.The model exhibits strong generalization and deterministic forecasting capabilities.

**Questions:**

Please refer to weaknesses.

**Ethical Concerns:**

["NO or VERY MINOR ethics concerns only"]

**Final Justification:**

From the authors response, there are still three points that make it difficult to be competitive.

1. There is no theoretical proofs for the bottleneck representations.

2. The strategy of "aggregates temporal and channel dependencies simultaneously" is proposed by "Unitst: Effectively modeling inter-series and intra-series dependencies for multivariate time series forecasting", which is not novel in this manuscript.

3. The experimental results show lower competitiveness, 57% experimental results are lower than baselines from the W2.

**Limitations:**

yes

**Quality:**

3

**Strengths And Weaknesses:**

Strengths:

1.Well-written and clearly presents its contributions. 2. Interesting approach, and experimental results are soundness. 3. The analysis and discussion are convincing.

Weaknesses:
1. The novelty is weak, since the latent bottleneck representations of encoder is introduced in time series forecasting, “Crossformer: Transformer Utilizing Cross-Dimension Dependency for Multivariate Time Series Forecasting” (ICLR2023). Also, there is no theoretical proofs for the bottleneck representation, please add. Therefore, the novelty is the learnable query-based decoder.
2. As a generalized forecasting model, some generalized baselines should compare, "One Fits All: Power General Time Series Analysis by Pretrained LM", "TOTEM: TOkenized Time Series EMbeddings for General Time Series Analysis" and "UniTS: A Unified Multi-Task Time Series Model"
3. In the learnable query of decoder, did the author consider adding covariates corresponding to the prediction target to enhance the expressiveness of the query?
4. Since the author states that TimePerceiver can encompasses extrapolation, interpolation, and imputation tasks along the temporal axis in lines 54-56, could the author do some comparisons within these tasks, "Optimal Transport for Time Series Imputation", "Csdi: Conditional score-based diffusion models for probabilistic time series imputation".

---

> ### Author Rebuttal · Authors · 2025-07-30
>
> Dear Reviewer pmxb,
>
> We sincerely thank you for your helpful feedback and insightful comments. In what follows, we address your concerns one by one. Note that we only report MSE metrics in this rebuttal due to its limited space.
>
> ---
>
> **[W1]** The novelty is weak, since the latent bottleneck representations of encoder is introduced in time series forecasting, “Crossformer: Transformer Utilizing Cross-Dimension Dependency for Multivariate Time Series Forecasting” (ICLR2023). Also, there is no theoretical proofs for the bottleneck representation, please add. Therefore, the novelty is the learnable query-based decoder.
>
> **[Response]** The use of latent bottleneck in TimePerceiver is fundamentally different from Crossformer [1] in the perspectives about (i) which information is aggregated into the latent and (ii) how to utilize the latent. First, our latent aggregates temporal and channel dependencies simultaneously via our latent bottleneck structure. This design explicitly captures interactions across both dimensions in a single operation. In contrast, Crossformer implicitly aggregates temporal-channel dependencies as interactions between temporal and channel information are considered separately. Here, Crossformer's bottleneck structure is used only in the cross-channel stage. Second, our latent is updated through multiple self-attention blocks. Such approach enables the latent to progressively refined through multiple self-attention blocks. The enhanced latent is then projected back to the input tokens, enabling it to incorporate more comprehensive and informative context. However, Crossformer's bottleneck structure is designed only for reducing computational cost, which is the reason why our TimePerceiver can outperform Crossformer.
>
> [1] Zhang et al., Crossformer: Transformer utilizing cross-dimension dependency for multivariate time series forecasting. ICLR 2023
>
> ---
>
> **[W2]** As a generalized forecasting model, some generalized baselines should compare, "One Fits All: Power General Time Series Analysis by Pretrained LM", "TOTEM: TOkenized Time Series EMbeddings for General Time Series Analysis" and "UniTS: A Unified Multi-Task Time Series Model".
>
> **[Response]** Thank you for the suggestion. We would like to clarify that our generalized formulation differs from the use of the term generalized in foundation models in that we redefine time-series forecasting itself to include extrapolation, interpolation, and imputation in a unified framework. In contrast, the three papers you mentioned (One Fits All, TOTEM, and UniTS) treat generalized as building a single foundation model that supports a wide variety of tasks, including classification, anomaly detection, and forecasting.
>
> Nevertheless, following your suggestion, we compare long-term forecasting results with the mentioned baselines with lookback window L=96. Note that “One Fits All” did not provide the results of L=96, we here omit the baseline. As shown below, our TimePerceiver outperforms TOTEM and is comparable to UniTS.
>
> | Datasets | TimePerceiver | TOTEM | UniTS |
> | :--- | :--- | :--- | :--- |
> | ETTh1 | 0.422 | 0.461 | 0.403 |
> | ETTh2 | 0.355 | 0.425 | 0.366 |
> | ETTm1 | 0.362 | 0.394 | 0.377 |
> | ETTm2 | 0.276 | 0.292 | 0.275 |
> | Weather | 0.238 | 0.239 | 0.235 |
> | ECL | 0.165 | 0.200 | 0.163 |
> | Traffic | 0.428 | 0.550 | 0.452 |
>
> ---
>
> **[W3]** In the learnable query of decoder, did the author consider adding covariates corresponding to the prediction target to enhance the expressiveness of the query?
>
> **[Response]** We constructed the query not merely as learnable parameters, but as a composition of covariates corresponding to the prediction target including temporal information (referred to TPE) and channel-specific information (referred to CPE). As shown below, experiments on the ECL dataset demonstrate that removing CPE, which encodes channel information, leads to a performance drop. This demonstrates that adding covariates corresponding to the prediction target enhances the expressiveness of the query, as you expected. We also experimented with adding an embedding for patch size (referred to PSE). However, since our trained model uses a fixed patch size, this addition had minimal impact.
>
> | Input Length | TPE \+ CPE (ours) | TPE | TPE \+ CPE \+ PSE |
> | :--- | :--- | :--- | :--- |
> | 96 | 0.125 | 0.135 | 0.127 |
> | 192 | 0.142 | 0.148 | 0.143 |
> | 336 | 0.161 | 0.166 | 0.161 |
> | 720 | 0.200 | 0.204 | 0.202 |
> | avg | 0.157 | 0.163 | 0.158 |
>
> ---
>
> **[W4]** Since the author states that TimePerceiver can encompasses extrapolation, interpolation, and imputation tasks along the temporal axis in lines 54-56, could the author do some comparisons within these tasks, "Optimal Transport for Time Series Imputation", "Csdi: Conditional score-based diffusion models for probabilistic time series imputation".
>
> **[Response]** Following your suggestion, we compare our framework with PSW-I [1] under the standard imputation setup [2,3] using 6:2:2 training/validation/test splits. As shown below, our TimePerceiver outperforms the baseline, PSW-I, under the ETTh1 dataset. We will add more comparisons (with other baselines, e.g., CSDI, and other datasets, e.g., Weather) into the final manuscript.
>
> | Dataset | TimePerceiver | PSW-I |
> | :---- | :---- | :---- |
> | ETTh1 | 0.218 | 0.253 |
>
> [1] Wang et al., Optimal Transport for Time Series Imputation, ICLR 2025 \
> [2] Wu et al., TimesNet: Temporal 2D-Variation Modeling for General Time Series Analysis, ICLR 2023 \
> [3] Goswami et al., MOMENT: A Family of Open Time-series Foundation Models, ICML 2024

---

> > ### Comment · Reviewer_pmxb · 2025-08-06
> > **Official Comment by Reviewer pmxb**
> >
> > Thanks for the authors insightful response. However, there are still three points that make it difficult to be competitive.
> >
> > 1. There is no theoretical proofs for the bottleneck representations.
> >
> > 2. The strategy of "aggregates temporal and channel dependencies simultaneously" is proposed by "Unitst: Effectively modeling inter-series and intra-series dependencies for multivariate time series forecasting", which is not novel in this manuscript.
> >
> > 3. The experimental results show lower competitiveness, 57% experimental results are lower than baselines from the W2.
> >
> > Thanks again for authors effort and time, I will keep my score.

---

> > > ### Author Response · Authors · 2025-08-06
> > > **Additional response to Reviewer pmxb**
> > >
> > > Dear Reviewer pmxb
> > >
> > > Thank you for the valuable observation. We appreciate the opportunity to clarify further.
> > >
> > > ---
> > >
> > > > There is no theoretical proofs for the bottleneck representations.
> > >
> > > We appreciate the reviewer’s comment. Although we agree that theoretical guarantees can further strengthen the contribution of a work, we respectfully believe that the lack of such proofs should not undervalue our paper. Many impactful works in the fields of deep learning and time-series forecasting have been recognized for their empirical effectiveness, even without formal proofs.
> > >
> > > We would like to emphasize that our latent bottleneck is extensively validated through extensive experiments (including additional experiments conducted during this rebuttal period) and shows consistent improvements across various benchmarks. We see theoretical analysis as a valuable future direction, and we thank the reviewer for highlighting this point.
> > >
> > > ---
> > >
> > > > The strategy of "aggregates temporal and channel dependencies simultaneously" is proposed by "Unitst: Effectively modeling inter-series and intra-series dependencies for multivariate time series forecasting", which is not novel in this manuscript.
> > >
> > > We respectfully disagree with the reviewer’s assessment regarding the novelty of our framework. First, as mentioned in our initial response, **our strategy is clearly distinct from UniTST, particularly in how the latent representations are utilized within the model.** In our framework, the latent serves as an **auxiliary memory** designed to capture diverse temporal-channel patterns. After aggregating all input tokens via cross-attention, we refine the latent memory through multiple self-attention blocks. **This design is conceptually and functionally different from UniTST**, while it is also capable of aggregating temporal and channel dependencies simultaneously.
> > >
> > > Second, we would like to emphasize that **our contribution is not limited to the ability to "aggregate temporal and channel dependencies simultaneously".** In our paper, we propose three components: (i) generalized formulation, (ii) latent bottleneck, and (iii) query-based decoder. These components are carefully designed to complement each other, and we validate both their individual impact and the strong forecasting performance across a wide range of benchmarks, as shown in Table 2 and Table 1 of the main manuscript, respectively. Taken together, we strongly believe that our paper makes several novel and impactful contributions.
> > >
> > > ---
> > >
> > > > The experimental results show lower competitiveness, 57% experimental results are lower than baselines from the W2.
> > >
> > > We would like to gently clarify this point. As shown below with standard deviations, among the seven benchmark datasets, our method (TimePerceiver) is meaningfully lower than UniTS in only one case (ETTh1), accounting for 14% of the results (not 57%). In contrast, we clearly outperform UniTS on three datasets (ETTh2, ETTm1, Traffic), and achieve comparable results on the remaining three (ETTm2, Weather, ECL).
> > >
> > > | Datasets | TimePerceiver | UniTS |
> > > | :---- | :---- | :---- |
> > > | ETTh1 | 0.422±0.005 | **0.403** |
> > > | ETTh2 | **0.355±0.007** | 0.366 |
> > > | ETTm1 | **0.362±0.003** | 0.377 |
> > > | ETTm2 | **0.276±0.002** | **0.275** |
> > > | Weather | **0.238±0.002** | **0.235** |
> > > | ECL | **0.165±0.003** | **0.163** |
> > > | Traffic | **0.428±0.008** | 0.452 |
> > >
> > > Therefore, we believe **it is fair to conclude that our method demonstrates stronger overall performance.** We appreciate the opportunity to clarify this and will include the comparison in the final manuscript.
> > >
> > > ---
> > >
> > > We sincerely appreciate your time and effort in engaging in constructive discussion to improve our work. If you still have any remaining concerns, we would be happy to address them further.
> > >
> > > Thank you very much, \
> > > Authors

---

> > > > ### Comment · Reviewer_QYcP · 2025-08-06
> > > >
> > > > In order to clear up ambiguities and ensure your comparisons are as rigorous as your efforts deserve, consider following Demšar (2006) by first performing a Friedman test and then applying a Wilcoxon signed rank post hoc test with Holm adjustment to control the family-wise error rate. You can use, e.g., scikit-posthocs, to do so. These results can be visualized with a Critical Difference diagram.
> > > >
> > > > However, this CD is hard to provide in this year's NeurIPS response format, the significance results can be provided in written form.
> > > >
> > > > Demšar, Janez. "Statistical comparisons of classifiers over multiple data sets." Journal of Machine learning research (2006): 1-30.

---

> > > > > ### Author Response · Authors · 2025-08-06
> > > > > **Additional response to Reviewer QYcP**
> > > > >
> > > > > Dear Reviewer QYcP,
> > > > >
> > > > > We appreciate the insightful suggestion.
> > > > >
> > > > > We found that the ranking-based statistical tests primarily focus on which method performs better or worse across datasets, rather than the magnitude of performance differences. In our case, for the three datasets (ETTm2, Weather, ECL), TimePerceiver is assigned a lower rank even though the actual differences are negligible. As a result, we observe that these tests conclude that TimePerceiver and UniTS perform similarly overall. In fact, TimePerceiver is meaningfully lower than UniTS in only one dataset (ETTh1), while outperforming UniTS in three others (ETTh2, ETTm1, Traffic). Therefore, we believe our original analysis remains valid: TimePerceiver demonstrates stronger overall performance.
> > > > >
> > > > > Moreover, the UniTS paper does not report standard deviations and the number of experimental runs, which makes a precise statistical comparison challenging. Therefore, we remain cautious about drawing definitive statistical conclusions at this stage. Nevertheless, we appreciate the suggestion to conduct such statistical comparisons and, if feasible or necessary, we will include them in the final manuscript.
> > > > >
> > > > > Thank you very much, \
> > > > > Authors

---

### Official Review · Reviewer_rg1y · 2025-07-02

**Clarity:** 3
**Significance:** 2
**Originality:** 2
**Rating:** 3
**Confidence:** 3

**Summary:**

The authors introduce a fresh approach to tackling a variety of time-series forecasting challenges—not just predicting the future, but also filling in gaps and making sense of missing data. Unlike older methods that mostly focus on just the encoder part of the model, TimePerceiver brings the encoder, decoder, and training process together into one unified system. Its design uses clever attention mechanisms to understand complex patterns across both time and different data channels, and the decoder is flexible enough to handle any target points you care about. Instead of relying on a two-step training process, TimePerceiver is trained end-to-end, which streamlines things and helps it perform better. The authors back up their claims with thorough experiments, showing that their model consistently outperforms other top methods. Overall, the paper makes a strong case for rethinking how we design and train models for time-series forecasting by taking a broader, more integrated approach.

**Questions:**

1. What are the computational costs (training time, memory footprint) of TIMEPERCEIVER compared to the baselines, especially given the dual attention mechanisms and bottleneck structure?

Suggestions: Please include a runtime and memory usage comparison across different sequence lengths. This is important to assess the trade-off between performance and scalability.

2. Under what conditions does TIMEPERCEIVER struggle? Are there failure cases or performance drop-offs in certain data regimes (e.g., small datasets, irregular sampling)?

Suggestions: Discuss known limitations and include a sensitivity analysis or case examples showing where the model might be less effective.

3. How does each architectural component (e.g., latent bottlenecks, learnable queries, self-attention in bottlenecks) individually contribute to performance?

Suggestions:  Please clearly  discuss each innovation's role. For example, what is the performance delta when removing learnable queries alone?

4. Can the authors elaborate on specific real-world scenarios where the simultaneous handling of extrapolation, interpolation, and imputation is essential?

Suggestions:  Add at least one case study or detailed scenario (e.g., in health, finance, or sensor networks) where this generalized forecasting capability provides a clear advantage over traditional models.

**Ethical Concerns:**

["Major Concern: Data privacy, copyright, and consent", "Major Concern: Safety and security", "Major Concern: Discrimination, bias, and fairness", "Major Concern: Environmental impact"]

**Limitations:**

The authors do not address the limitations of the proposed framework. While the technical contributions are clearly presented, the following points  should be addressed:

a) The paper should explicitly discuss scenarios where TIMEPERCEIVER might fail or underperform, such as:  sparse time-series data, extremely long sequences where attention-based architectures struggle with memory and speed.

b) The authors should comment on the training and inference costs. High computational demands can have environmental and accessibility implications.

c) If deployed in domains like health or finance, biases in training data could propagate through the model. The authors should acknowledge these risks and suggest monitoring or mitigation strategies.

d) If used in sensitive settings (e.g., medical records, financial logs), how does the model handle privacy risks, particularly if used for imputation of missing data?

**Quality:**

3

**Strengths And Weaknesses:**

This authors present how it brings together different parts of time-series forecasting—encoding, decoding, and training—into one well-designed, unified approach. The authors use attention mechanisms and bottleneck representations to help the model capture complex patterns, and they back up their claims with thorough experiments and diagrams that make even the vague concepts easier to understand. The authors expand the scope of forecasting to include not just predicting the future, but also filling in gaps and handling missing data, making their method useful for a wide range of real-world problems. The way the authors combine existing ideas in a new way shows real innovation and could have an impact across fields like health, finance, and climate science.

On the other hand, the model’s complexity could be hard to use in situations where resources are limited or real-time results are needed, and the paper doesn’t discuss these practical challenges. While the framework’s generality is a good idea, it would be more convincing if the authors gave more examples of where all these forecasting tasks are needed in the real world. Lastly, many of the individual building blocks come from other areas, so the true originality is in the combination rather than the components themselves.

---

> ### Author Rebuttal · Authors · 2025-07-30
>
> Dear Reviewer rg1y,
>
> We sincerely thank you for your helpful feedback and insightful comments. In what follows, we address your concerns one by one. Note that we only report MSE metrics in this rebuttal due to its limited space.
>
> ---
>
> **[Q1]** What are the computational costs (training time, memory footprint) of TIMEPERCEIVER compared to the baselines, especially given the dual attention mechanisms and bottleneck structure? Please include a runtime and memory usage comparison across different sequence lengths.
>
> **[Response]** As shown below, TimePerceiver demonstrates efficiency in both training time and memory footprint compared to baselines, regardless of the sequence length. This efficiency stems from the latent bottleneck that enables compression of core information efficiently. Additional analyses, such as the impact of longer prediction lengths, are provided in Appendix B.
>
> | Models | Seq Length | Memory(MB) | Runtime(s/iter) | Models | Seq Length | Memory(MB) | Runtime(s/iter) |
> | :---- | :---- | :---- | :---- | :---- | :---- | :---- | :---- |
> | TimePerceiver | 96 | 660 | 0.0435 | PatchTST | 96 | 1512 | 0.0547 |
> |  | 384 | 918 | 0.0583 |  | 384 | 6852 | 0.1156 |
> | DeformableTST | 96 | 972 | 0.0919 | Crossformer | 96 | 2768 | 0.1299 |
> |  | 384 | 2998 | 0.1153 |  | 384 | 4110 | 0.1350 |
> | CARD | 96 | 538 | 0.0737 | Timesnet | 96 | 1300 | 0.1376 |
> |  | 384 | 724 | 0.0828 |  | 384 | 3258 | 0.1385 |
>
> ---
>
> **[Q2]** Under what conditions does TIMEPERCEIVER struggle? Are there failure cases or performance drop-offs in certain data regimes (e.g., small datasets, irregular sampling)? Discuss known limitations and include a sensitivity analysis or case examples showing where the model might be less effective.
>
> **[Response]** TimePerceiver is designed to be effective even in small datasets, as our generalized formulation enables the model to learn from a richer context beyond standard future prediction, especially by leveraging extrapolation, interpolation and imputation. However, in cases where the available time-series data is extremely short, the benefits of the generalized formulation may become less pronounced compared to the standard formulation. However, even in such cases, we observe that performance remains comparable to that of standard formulation.
>
> ---
>
> **[Q3]** How does each architectural component (e.g., latent bottlenecks, learnable queries, self-attention in bottlenecks) individually contribute to performance? Please clearly discuss each innovation's role. For example, what is the performance delta when removing learnable queries alone?
>
> **[Response]** We would like to note that the analysis of each component is discussed in detail in Section 4.2 component ablation studies and Appendix D. Specifically, in the analysis, we conducted experiments by replacing each architectural component of TimePerceiver with a different structure or modifying the model formulation to verify the effectiveness of each component. We observe that the latent bottleneck contributed to performance improvement by efficiently aggregating temporal-channel dependencies. Additionally, the combination of learnable queries and a generalized formulation led to performance improvements due to its ability to handle diverse temporal contexts. We here further provide additional experiments with varying the number of latent tokens. As shown below, the results show a clear performance drop without latent bottleneck, highlighting its role in effectively modeling temporal and cross-channel dependencies.
>
> | Latent Size | ECL | Traffic |
> | :--- | :--- | :--- |
> | 0 | 0.165 | 0.415 |
> | 8 | 0.158 | 0.401 |
> | 16 | 0.159 | 0.404 |
> | 32 | 0.158 | 0.405 |
>
> We would like to clarify that our generalized formulation cannot be implemented without our query-based decoder, so we did not provide the ablation study for our query-based decoder.
>
> ---
>
> **[Q4]** Can the authors elaborate on specific real-world scenarios where the simultaneous handling of extrapolation, interpolation, and imputation is essential? Add at least one case study or detailed scenario (e.g., in health, finance, or sensor networks) where this generalized forecasting capability provides a clear advantage over traditional models.
>
> **[Response]** A concrete example is climate and environmental monitoring using sensor networks that measure variables such as temperature, humidity, and air quality. In such systems, our generalized forecasting capability is essential. First, imputation or interpolation is needed when sensor failures or communication issues lead to missing data. Second, extrapolation enables the estimation of past values from periods before sensor deployment. Lastly, forecasting is required to predict future conditions for effective planning and decision-making.
>
> We believe our unified framework, built upon our generalized formulation, can flexibly support all of these objectives, making it particularly well-suited for real-world applications like environmental monitoring. We will incorporate this discussion into the final manuscript.
>
> ---
> **[Limitations]** Thank you for the insightful comments. We provide additional responses regarding the mentioned limitations below. We also would like to note that we discussed limitations in Appendix E of the supplementary material.
>
> **[a]** Please refer to [Q2]. \
> **[b]** Please refer to [Q1]. \
> **[c]** We believe our framework is well-suited to address these biases, thanks to the generalized formulation that enables diverse contextual understanding and the latent bottleneck that effectively captures various temporal-channel dependencies. \
> **[d]** Thanks to the the efficiency of our framework as shown in [Q1], we believe that it is executable in resource-limited edge devices without network access (e.g., uploading/downloading private data). This data isolation could mitigate such privacy risks.

---

> > ### Author Response · Authors · 2025-08-07
> > **A gentle reminder to Reviewer rg1y**
> >
> > Dear Reviewer rg1y,
> >
> > We hope this message finds you well. As **there are only two days remaining in the Author-Reviewer Discussion period**, we would like to kindly follow up on our rebuttal.
> >
> > We have made a sincere effort to address your concerns through detailed clarifications, additional experiments, and further analysis. We believe these additions meaningfully strengthen our paper and help clarify important points.
> >
> > If there are any remaining concerns or unresolved issues, please don’t hesitate to let us know. We would be happy to provide further clarification.
> >
> > Thank you once again for your time and valuable feedback.
> >
> > Best regards, \
> > Authors

---

### Official Review · Reviewer_DDKH · 2025-07-03

**Clarity:** 4
**Significance:** 3
**Originality:** 2
**Rating:** 5
**Confidence:** 3

**Summary:**

This paper presents **TimePerceiver**, a unified encoder-decoder framework for multivariate time series forecasting. The core idea is to generalise the training task beyond standard extrapolation tasks to include **interpolation** and **imputation** for arbitrary missing input.

To that end, the authors propose a Transformer-based model using patch-based tokenisation and cross-variant attention. First, the context patches are transformed into a token representation augmented with temporal (patch position) and channel (index of dimension) information. The encoder compresses context patches into latent tokens (with size and dimensions to be chosen) and reprojects the contextualised input tokens into the original token space. The decoder uses tokens only based on temporal and channel information and uses cross-attention to transform these tokens into output tokens that are projected back into signal space.

The model is evaluated across 8 datasets with multiple forecasting horizons and consistently demonstrates strong performance, often achieving state-of-the-art or near-best results, with favourable computational and parameter efficiency. Furthermore, the authors show through ablation studies the benefits of their choices and show that the seasonality is recovered through the cross attention patterns.

**Questions:**

- The decoder operates by projecting contextualised query tokens back into full patch outputs. It remains unclear whether bypassing re-expansion to full resolution (i.e., using latent tokens directly) might yield a more efficient architecture. Have the authors tried directly using the encoded latent token as input for the decoder? If so, how much was the performance degrading?
- I am not sure I understand the difference between interpolation and imputation in the description of the project.
- I am not fully convinced that this framework would be able to directly handle arbitrary time segments as is, as this would blur the captured seasonality. I think additional information on the size of the patch should be integrated into the representation, perhaps through the same system as temporal and covariate index information. Have the authors tried anything in that regard?

**Ethical Concerns:**

["NO or VERY MINOR ethics concerns only"]

**Final Justification:**

The original submission was already quite convincing, presenting a simple yet effective architecture that leverages a latent bottleneck to enable efficient computation. The additional experiments have thoroughly addressed all of my concerns:
- The inclusion of standard deviation demonstrates the statistical robustness of the results.
- The analysis of latent size (extended in the response to Reviewer QYcP) and the "reprojection layer" provides sound justification for the architectural choices.
- The additional experiments on imputation show competitive results compared to other baselines.
- The additional details on training time and model complexity given to Reviewer rg1y make a compelling point for the benefits of the method despite not reaching SOTA on every dataset.

I have increased my score in light of the authors' additional experiments, provided they are included in the final version.

**Limitations:**

The limitations of the models are correctly addressed but the statistical significance could be improved by including the standard deviation of the 5 runs of the experiments.

**Quality:**

4

**Strengths And Weaknesses:**

- **Originality**:  While many of the ideas exist in other works: related ideas for the pretraining (e.g., masking-based pretraining) ([1,2]) and of having per channel and per patch attention by explicitely tokenizing them individually [3],  to the best of my knowledge, this is the first model not doing it in a two steps setting while limiting the computation and number of parameters needed. Thus, the learnable latent bottleneck is an interesting addition.

- **Quality**:
  - (+)The model is built on clear motivations, and the experimental results are convincing.
  - (-) Since the results were averaged over 5 runs, it could have been interesting to show the standard deviation of the results to show the statistical significance of the results (though it is not standard in the field).
  - (+) The ablation study is compelling to experimentally show the benefits of their methods.
  - (+) The visual analysis of the cross-attention map recovering the seasonality of the TS is an interesting addition.
  - (-) Multiple latent sizes are tested, I suppose the choice is based on performance, but the procedure is not detailed. Studying the sensitivity of the model to this parameter should be included in the study. Furthermore, the reprojection in the original size is not motivated either.
  - (-) Since the method was trained on an extended task able to handle imputation and interpolation, it could have been interesting to evaluate the performance on such tasks.


- **Clarity**:
    The paper is well-structured with intuitive visualisations (e.g., Figures 1 and 2) that effectively explain the generalised forecasting setup and encoder-decoder flow.

- **Significance**:  Though many ideas seem to have been around already, this architecture with a learnable latent bottleneck allowing faster computation seems of interest for the community. Furthermore, the simplicity of the architecture, the very compelling results on different datasets and the superiority of the method against many SOTA methods make this contribution significant.


    [1]|Dong, J., Wu, H., Zhang, H., Zhang, L., Wang, J., & Long, M. (2023). Simmtm: A simple pre-training framework for masked time-series modeling. _Advances in Neural Information Processing Systems_, _36_, 29996-30025.

    [2] Kollovieh, M., Ansari, A. F., Bohlke-Schneider, M., Zschiegner, J., Wang, H., & Wang, Y. B. (2023). Predict, refine, synthesize: Self-guiding diffusion models for probabilistic time series forecasting. _Advances in Neural Information Processing Systems_, _36_, 28341-28364.

    [3] Zhang, Y., & Yan, J. (2023, May). Crossformer: Transformer utilizing cross-dimension dependency for multivariate time series forecasting. In _The eleventh international conference on learning representations_.

---

> ### Author Rebuttal · Authors · 2025-07-30
>
> Dear Reviewer DDKH,
>
> We sincerely thank you for your helpful feedback and insightful comments. In what follows, we address your concerns one by one. Note that we only report MSE metrics in this rebuttal due to its limited space.
>
> ---
>
> **[W1]** Since the results were averaged over 5 runs, it could have been interesting to show the standard deviation of the results to show the statistical significance of the results (though it is not standard in the field).
>
> **[Response]** Following your suggestion, we evaluate the stability of TimePerceiver by reporting mean and standard deviation over five runs with lookback window L=384. As shown below, across all 32 experimental settings, 25 cases exhibit a standard deviation less than or equal to 0.0005, indicating that TimePerceiver remains highly stable.
>
> | Input Length | ETTh1 | ETTh2 | ETTm1 | ETTm2 |
> | :---- | :---- | :---- | :---- | :---- |
> | 96 | 0.365±0.004 | 0.275±0.004 | 0.284±0.007 | 0.160±0.005 |
> | 192 | 0.397±0.002 | 0.335±0.003 | 0.323±0.004 | 0.215±0.003 |
> | 336 | 0.412±0.006 | 0.344±0.009 | 0.353±0.004 | 0.276±0.003 |
> | 720 | 0.442±0.008 | 0.392±0.010 | 0.399±0.002 | 0.359±0.006 |
> | **Input Length** | **Weather** | **Solar-Energy** | **ECL** | **Traffic** |
> | 96 | 0.142±0.002 | 0.161±0.006 | 0.124±0.001 | 0.374±0.001 |
> | 192 | 0.190±0.003 | 0.178±0.003 | 0.141±0.002 | 0.381±0.002 |
> | 336 | 0.244±0.000 | 0.188±0.003 | 0.161±0.003 | 0.396±0.002 |
> | 720 | 0.313±0.003 | 0.195±0.005 | 0.202±0.002 | 0.431±0.003 |
>
> ---
>
> **[W2]** Multiple latent sizes are tested, I suppose the choice is based on performance, but the procedure is not detailed. Studying the sensitivity of the model to this parameter should be included in the study.
>
> **[Response]** Thank you for mentioning the important point. In our experiments, we simply chose the latent size based on validation. Following your suggestion, we further explore its sensitivity as reported below. We observe that the presence of a latent bottleneck clearly improves performance over having none, but the specific number of latent tokens plays only a minor role in determining the final performance, as they are all relatively small compared to the number of input tokens. This indicates that our model is not overly sensitive to this parameter, and even with a small latent size, we observe substantial performance gains.
>
> | Latent Size | ECL | Traffic |
> | :--- | :--- | :--- |
> | 0 | 0.165 | 0.415 |
> | 8 | 0.158 | 0.401 |
> | 16 | 0.159 | 0.404 |
> | 32 | 0.158 | 0.405 |
>
> ---
>
> **[W3]** Since the method was trained on an extended task able to handle imputation and interpolation, it could have been interesting to evaluate the performance on such tasks.
>
> **[Response]** Following your suggestion, we conduct imputation experiments with three datasets (ETTh1, Weather, Electricity) and three baselines (TimesNet [1], iTransformer [2], PatchTST [3]). We here set a more challenging task of imputation via patch-wise masking with a patch length of 24, as opposed to the easier setup of masking at individual timesteps. As shown below, our TimePerceiver consistently outperforms the baselines across all datasets. We will add these results with more baselines into the final manuscript.
>
> | Dataset | Masked Patch | TimePerceiver | TimesNet | iTransformer | PatchTST |
> | :---- | :---- | :---- | :---- | :---- | :---- |
> | ETTh1 | 1 | 0.218 | 0.301 | 0.257 | 0.241 |
> |  | 2 | 0.248 | 0.309 | 0.267 | 0.266 |
> | Weather | 1 | 0.066 | 0.072 | 0.066 | 0.061 |
> |  | 2 | 0.078 | 0.090 | 0.073 | 0.069 |
> | Electricity | 1 | 0.097 | 0.120 | 0.093 | 0.145 |
> |  | 2 | 0.106 | 0.123 | 0.102 | 0.156 |
>
>
> [1] Wu et al., TimesNet: Temporal 2D-Variation Modeling for General Time Series Analysis, ICLR 2023 \
> [2] Liu et al., iTransformer: Inverted Transformers Are Effective for Time Series Forecasting, ICLR 2024 \
> [3] Nie et al., A Time Series is Worth 64 Words: Long-term Forecasting with Transformers, ICLR 2023
>
> ---
>
> **[W2]** Furthermore, the reprojection in the original size is not motivated either.\
> **[Q1]** The decoder operates by projecting contextualised query tokens back into full patch outputs. It remains unclear whether bypassing re-expansion to full resolution (i.e., using latent tokens directly) might yield a more efficient architecture. Have the authors tried directly using the encoded latent token as input for the decoder? If so, how much was the performance degrading?
>
> **[Response]** Thank you for the thoughtful comment. We have conducted experiments with the variant (referred to as DirectLT) you suggested in which the decoder uses latent tokens directly without re-expansion to full resolution. As shown below, our decoder design is significantly better than directly using the encoded latent token as input for the decoder.
>
> We believe the superiority of our design stems from the following insight. In our framework, the latent bottleneck is designed to capture global temporal-channel patterns, while the original input sequence retains fine-grained local signals. Relying solely on the latent makes it difficult to reconstruct these detailed signals, ultimately leading to performance degradation. In contrast, our framework enriches input patch representations by leveraging the latent, enabling each to incorporate both global contexts and local details.
>
> | Input Length | Ours | DirectLT |
> | :--- | :--- | :--- |
> | 96 | 0.366 | 0.423 |
> | 192 | 0.394 | 0.437 |
> | 336 | 0.413 | 0.429 |
> | 720 | 0.445 | 0.473 |
> | Avg | 0.404 | 0.440 |
>
> ---
>
> **[Q2]** I am not sure I understand the difference between interpolation and imputation in the description of the project.
>
> **[Response]** Your understanding is generally correct. Interpolation can indeed be considered a subset of imputation. However, we made the distinction because interpolation refers to filling in values within continuous gaps, while imputation includes both such cases and those where individual points are missing. As you insightfully pointed out, our model does not explicitly separate these cases, but rather handles them within a generalized formulation.
>
> ---
>
> **[Q3-1]** I am not fully convinced that this framework would be able to directly handle arbitrary time segments as is, as this would blur the captured seasonality.
>
> **[Response]** We gracefully disagree with the concern that our framework may hinder the model’s ability to capture seasonality. On the contrary, we believe that the ability to handle arbitrary time segments enables the model to learn a broader range of recurring seasonal patterns.
>
> Our generalized formulation allows the model to learn from a wider range of configurations such as predicting the present given both past and future, or inferring the past from the future alone. By exposing the model to such diverse temporal contexts, we enhance rather than blur its ability to capture seasonal patterns. Crucially, latent tokens aggregate relationships among patches across every temporal‑channel tokens, creating a compressed global summary that captures seasonal patterns. This perspective is supported by our empirical results, where the generalized formulation consistently outperforms standard formulation across multiple settings.
>
> ---
>
> **[Q3-2]**  I think additional information on the size of the patch should be integrated into the representation, perhaps through the same system as temporal and covariate index information. Have the authors tried anything in that regard?
>
> **[Response]** Regarding your suggestion to provide additional information on the size of the patch, since the current TimePerceiver implementation is trained with a fixed patch length, we did not include patch size embeddings in our framework. As shown below with our experiments on ETTh1, we inject a patch length embedding into both inputs and queries that produced no measurable gains in our ablation study. Therefore such a patch-size embedding has not proved necessary for the current setting.
>
> Instead, based on your idea, we further validate a new multi-resolution design with patch-size embeddings: (i) we patchify the input time-series with various patch sizes 12, 24, 48, and then (ii) use all patches as input tokens in our framework. As shown below, it has not yet delivered a clear performance lift.
>
> However, your insightful suggestion will be strongly considered in the future when developing time-series foundation models, especially when combined with our generalized formulation, since we believe that enabling the model to process time-series at multiple resolutions is critical for scaling it to handle large-scale data.
>
> | Input Length | TimePerceiver (Ours) | TimePerceiver (Patch Size Embedding) | TimePerceiver (Multi-Resolution) |
> | :--- | :--- | :--- | :--- |
> | 96 | 0.366 | 0.366 | 0.364 |
> | 192 | 0.394 | 0.390 | 0.395 |
> | 336 | 0.413 | 0.417 | 0.414 |
> | 720 | 0.445 | 0.445 | 0.473 |
> | Avg | 0.404 | 0.405 | 0.412 |

---

> > ### Comment · Reviewer_QYcP · 2025-08-04
> > **Question regarding W3**
> >
> > I have a question regarding your response to W3. You mention "TimePerceiver consistently outperforms the baselines across all datasets." As far as I see, you only outperform the other methods on a single dataset (ETTh1), or do I observe the results in the wrong way?
> >
> > Thank you for your clarifications.

---

> ### Author Response · Authors · 2025-08-05
> **Response regarding W3**
>
> Dear Reviewer QYcP,
>
> Thank you for your comments and for taking the time to review our manuscript.
>
> For **W3**, we apologize for the confusion. Our intention was to emphasize that **TimePerceiver consistently achieves strong performance across all datasets**, not necessarily that it outperforms all baselines on every single one.
>
> In addition, it is worth noting that, even though we did not specifically tailor our framework to the imputation task, we were still able to achieve competitive results as reported in the table.
>
> Please do not hesitate to let us know if there are any remaining concerns or clarifications you would like us to address. We are more than willing to make the most of the remaining discussion time to further improve our paper.
>
> Thank you very much, \
> Authors

---

> > ### Comment · Reviewer_QYcP · 2025-08-05
> >
> > Thank you. I think the results are strong, I would just change the wording slightly and call it competitive not outperforming.

---

> ### Comment · Reviewer_DDKH · 2025-08-06
>
> I want to thank the authors for the time and effort invested in conducting additional experiments to address the raised concerns. The new results help clarify the strengths and design choices of the proposed method.
>
> While the imputation performance does not surpass all baselines, it remains competitive—particularly given the advantages in training efficiency and memory usage compared to existing methods (as noted in the response to Reviewer rg1y).
>
> I also found the analysis of latent size (extended in the answer to reviewer QYcP) particularly insightful. Though surprising, the observation that performance saturates with as few as 8 latent tokens supports the authors’ intuition in [W2-Q1], namely that the latent tokens primarily capture global channel-level patterns, while the reprojection helps recover local details.
>
> All of my questions and weaknesses have been addressed; provided the additional experiments are included in the appendix for the final version, this paper would make a strong addition to the field of time series forecasting. I raised my score and would recommend this paper for publication.

---

> > ### Author Response · Authors · 2025-08-06
> >
> > Dear Reviewer DDKH,
> >
> > Thank you very much for your thoughtful comments and for taking the time to review our manuscript. We are glad that we were able to address and improve upon the points you suggested, including the necessary aspects of our method and further directions we could try. We will incorporate all the findings and the discussions from the rebuttal into the final manuscript. We hope that our method will be further extended in the field of time series domain.
> >
> > If you have any further suggestions, we would be happy to hear them.
> >
> > Sincerely,\
> >  Authors

---

> ### Comment · Area_Chair_QjWq · 2025-08-06
> **Mandatory Acknowledgement**
>
> Thanks for the active engagement in the discussion with the authors and filling in the final justification.
>
> Please remember to also submit the mandatory acknowledgement as well.
>
>
>
> Thanks,
>
> -Area Chair

---

### Author Response · Authors · 2025-08-03
**A Gentle Reminder to AC and Reviewers**

Dear AC and Reviewers,

We hope this message finds you well. We are writing to kindly follow up regarding our rebuttal.

We have made a sincere effort to address the reviewers' concerns through detailed clarifications, additional experiments, and further analysis. We believe these additions meaningfully strengthen our paper and help clarify important points.

If there are any remaining concerns or unresolved issues, please feel free to let us know. We would be happy to provide further clarification.

Thank you again for your time and valuable feedback.

Best regards, \
Authors

---

### Decision · Program_Chairs · 2025-09-17

**Decision:**

Accept (poster)

**Comment:**

**(a) Scientific claims and findings.**
The paper proposes TimePerceiver, a generalized encoder–decoder framework for time-series forecasting that unifies extrapolation, interpolation, and imputation within one architecture. It introduces a latent bottleneck representation and learnable decoder queries, demonstrating strong empirical results across multiple benchmarks.

**(b) Strengths.**
The work is well-motivated, clearly presented, and empirically validated with extensive experiments and ablations. Its unified formulation and bottleneck design show efficiency and adaptability, and the results are competitive or state-of-the-art across diverse datasets.

**(c) Weaknesses.**
Concerns remain about limited novelty relative to prior work, lack of theoretical justification for the bottleneck, incomplete coverage of recent baselines, and insufficient discussion of limitations such as scalability, real-world deployment, and statistical robustness.

**(d) Decision rationale**
Despite reservations about originality and missing baselines, the unified framework, convincing empirical performance, and thorough responses during the rebuttal collectively justify acceptance. The contribution is likely to be impactful for the time-series forecasting community.  Acceptance is recommended contingent on including the additional comparisons, analyses, and clarifications promised in the discussion.

**(e) Discussion and rebuttal.**
Reviewers initially raised concerns about novelty, missing baselines, computational costs, and lack of clarity in definitions and significance.  The authors addressed these by adding efficiency analyses, imputation experiments, ablation details, comparisons with stronger baselines, and case studies illustrating real-world use.  While some novelty and theoretical questions persist, the responses and planned revisions were sufficient to alleviate most concerns.  On balance, the paper benefits from the discussion and should be acceptable provided improvements are incorporated.